# Capping protein-controlled actin polymerization shapes lipid membranes

Katharina Dürre[1], Felix C. Keber[1,2], Philip Bleicher[1], Fridtjof Brauns[3], Christian J. Cyron[4], Jan Faix [5] &
Andreas R. Bausch[1]

Arp2/3 complex-mediated actin assembly at cell membranes drives the formation of protrusions or endocytic vesicles. To identify the mechanism by which different membrane deformations can be achieved, we reconstitute the basic membrane deformation modes of inward and outward bending in a confined geometry by encapsulating a minimal set of cytoskeletal proteins into giant unilamellar vesicles. Formation of membrane protrusions is favoured at low capping protein (CP) concentrations, whereas the formation of negatively bent domains is promoted at high CP concentrations. Addition of non-muscle myosin II results in full fission events in the vesicle system. The different deformation modes are rationalized by simulations of the underlying transient nature of the reaction kinetics. The relevance of the regulatory mechanism is supported by CP overexpression in mouse melanoma B16-F1 cells and therefore demonstrates the importance of the quantitative understanding of microscopic kinetic balances to address the diverse functionality of the cytoskeleton.

---

[1] Lehrstuhl für Zellbiophysik, Technische Universität München, James-Franck-Str. 1, D-85748 Garching, Germany. [2] Lewis-Sigler Institute for Integrative Genomics, Princeton University, Princeton, NJ 08544, USA. [3] Arnold Sommerfeld Center for Theoretical Physics and Center for NanoScience, Department of Physics, Ludwig-Maximilians-Universität München, Theresienstraße 37, D-80333 München, Germany. [4] Lehrstuhl für Numerische Mechanik, Technische Universität München, Boltzmannstraße 15, D-85748 Garching, Germany. [5] Institut für Biophysikalische Chemie, Medizinische Hochschule Hannover, Carl-Neuberg Straße 1, D-30625 Hannover, Germany. Correspondence and requests for materials should be addressed to A.R.B. (email: abausch@mytum.de)

For actin, most, if not all of de novo nucleation appears at cell membranes resulting in characteristic self-organization effects, which can lead to the formation of filopodia or lamellipodia[1–11]. One key player is the nucleating Arp2/3 complex, which initiates the formation of the dendritic-like, branched networks at the plasma membrane and leads to the propulsion of spherical objects[12–16] in absence of any motor proteins. ATP hydrolysis driven filament polymerization against the plasma membrane is sufficient to push the membrane forward. Thereby the resulting membrane growth speed depends on the number of pushing actin filaments[17,18]. The biochemical properties of the Arp2/3 complex have been well characterized[19,20]. The Arp2/3 complex is activated by VCA (verprolin, central, acidic) domains of N-WASP (Wiskott–Aldrich syndrome protein) and SCAR/ WAVE (suppressor of cAMP receptor/WASP-family verprolin-homologous protein) proteins at the plasma membrane to create new branches from preexisting mother filaments. Coupled to the outside of a lipid vesicle, such actin networks can create sufficient forces to produce protrusions into the vesicle by self-stabilization of the membrane[21]. Surprisingly, the very same nucleation and growth processes are also thought to be responsible for the formation of negative curvatures in the plasma membrane and final membrane fission, as found in endocytosis[22–25]. However, still very little is known about the processes that allow cells to switch between these two fundamentally different deformation modes. Clearly, the microscopic organization of the actin cytoskeleton defines its function, yet how it is achieved remains to be resolved. A central regulator of the microscopic network properties is heterodimeric capping protein (CP)[15,26–29]. It binds to actin filament barbed ends to inhibit polymerization[30]. In the presence of CP branching by Arp2/3 complex is favoured[15], whereas depletion of CP results in the formation of actin bundles and thus switches from lamellipodium to filopodium formation[2,7,29]. Local monomer depletion is a second parameter, which controls network density and architecture. The local depletion leads to different growth velocities of the polymerizing actin network and controls steering of the actin network[31].

To study the physical principles underlying membrane deformation in vitro reconstitution systems are a promising approach[21,32–37]. By reconstitution of actin cortices inside of giant unilamellar vesicles (GUVs) it was shown that only contractile actin cortices are capable to drive shape remodelling and blebbing[36,37], while in the absence of any molecular motors no shape deformation of the membrane could be observed[35].

In the present study, we encapsulated actin, Arp2/3 complex and CP inside of GUVs and localized Arp2/3 complex activation to the membrane by coupling His-tagged VCA to nickel-nitrilotriacetic acid (Ni-NTA) lipids. We show that a limited pool of actin monomers can lead to inhomogeneous actin polymerization, where CP regulates the local actin density at the membrane. We found two distinct network architectures, which deformed the membrane either outwards or inwards. Only at high CP concentrations and in presence of the molecular motor non-muscle myosin II (NMM II) membrane invaginations were induced, which ultimately led to fission events. Simulating the underlying reaction diffusion kinetics under depleting protein pool conditions revealed that two distinct network patterns of actin growth beneath the membrane could be distinguished. The resulting inhomogeneous growth velocity profile of the network promoted the formation of membrane protrusions at low CP concentration and induced negative membrane curvatures at high CP concentration. Overexpression of CP in B16-F1 cells supported the role of CP in switching between these two deformation modes, since besides diminished lamellipodium formation and inhibition of microspikes, increased CP concentration markedly amplified the number of intracellular vesicles. Altogether, CP

controls membrane localized growth kinetics of Arp2/3 complex-mediated actin polymerization and thereby determines membrane shape deformations.

## Results

**Arp2/3 complex-induced polymerization at the membrane.** We manufactured a minimal model system of a cytoskeletal vesicle by encapsulating actin monomers and actin-binding proteins into vesicles with a radius of 10–25 μm using a modified emulsion transfer technique continuous droplet interface crossing encapsulation (cDICE) (Fig. 1a)[37,38]. The lipid membrane was composed of a mixture of Egg-PC, PEG-PE and Ni-NTA. The encapsulation of actin monomers alone resulted in the spontaneous nucleation and polymerization of actin filaments spanning the vesicle's volume without any enrichment of the network at the membrane (Fig. 1b). In the following series of experiments we added Arp2/3 complex, His-tagged VCA from N-WASP and profilin to the solution prior to encapsulation. Profilin blocks polymerization at the pointed end of actin filaments and diminishes the spontaneous actin nucleation in the vesicle's volume. His-tagged VCA was tethered to the membrane by Ni-NTA lipids to restrict Arp2/3 complex-mediated actin polymerization to the membrane resulting in localized networks due to the self-amplifying nature of Arp2/3 complex. Still a significant amount of actin was polymerized spontaneously, resulting in a fine volume spanning network (Fig. 1c). This is consistent with measurements from bulk actin-pyrene assays, which showed that despite the presence of profilin actin assembly was instantaneously initiated due to high amounts of VCA-activated Arp2/3 complex (Fig. 1d).

**CP is the central component for localized network formation.** CP binds to actin filament barbed ends and in cooperation with profilin suppresses spontaneous nucleation and elongation of filaments in solution as seen in actin-pyrene fluorescence measurements (Supplementary Figure 1). Measurements of pyrene fluorescence of the full system in the presence of 20 nM CP thus showed a lag phase of $1.5 \pm 0.3$ min in the polymerization curve (Fig. 1d). During encapsulation in vesicles this lag phase should ensure that the Arp2/3 complex can freely diffuse to the membrane before its activation as nucleation of actin was effectively suppressed within the volume of the vesicles. Indeed the encapsulation of this system into vesicles resulted for 78% of the vesicles in the localization of actin networks at the membrane without appearance of any visible volume spanning network. Two types of network morphologies could be distinguished (Fig. 1e). In 36% of the vesicles, spatially separated actin patches (domains) appeared at the membrane (Supplementary Figure 2). In 42% of the vesicles, almost homogeneous closed cortices were observed. This suggests that small fluctuations of CP concentrations are already sufficient to modify nucleation rates leading to the formation different domain morphologies.

A further increase of CP resulted in the formation of three distinct domain morphologies of actin, which could be distinguished based on their curvature: (i) convex actin protrusions (protrusion) with a length of up to 4 μm (Fig. 2a). These were able to push forward the vesicle's membrane and induced a convex deformation of the membrane (Supplementary Figure 3). (ii) Concavely bent actin domains (concave domain), which induced a localized negative curvature into the vesicle's membrane (Fig. 2b and Supplementary Figure 3) and (iii) flat actin domains, which had no discernible change of membrane curvature (Fig. 2c). The appearance and distribution of the different vesicle morphologies depended on the CP concentration (Fig. 2d). While the vesicles with a protrusion configuration almost exclusively appeared at 40

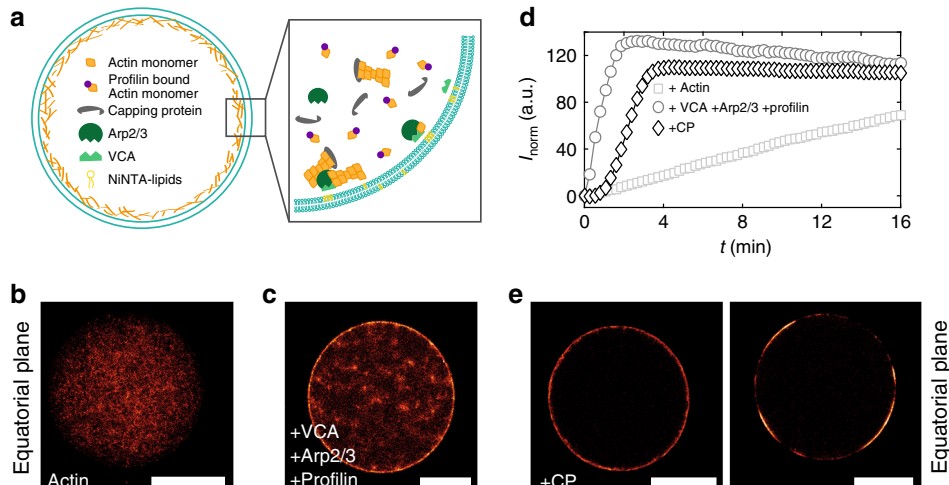

**Fig. 1** Actin polymerization at the membrane is mediated by Arp2/3 complex and CP. **a** Molecular components required to produce cytoskeletal vesicles: Actin, His-tagged VCA, Arp2/3 complex, CP and profilin were encapsulated inside of giant unilamellar vesicles (GUVs). They contained 10% Ni-NTA lipids to localize His tagged VCA to the membrane and 2.5% PEG2000 PE to avoid unspecific protein-membrane interactions. **b**, **c** Confocal images of the equatorial plane of the cytoskeletal vesicles are shown. The encapsulation of only 3 μM actin (red hot) resulted in pronounced bulk polymerization (**b**). The addition of 0.3 μM Arp2/3, 0.3 μM His-tagged VCA and 13.5 μM profilin to the solution prior to encapsulation induced localized actin polymerization at the membrane (**c**). **d** Pyrene fluorescence assays of actin polymerization in the dependence of actin-binding proteins (ABPs): The polymerization rate of 3 μM actin (squares) was increased upon addition of 0.3 μM VCA, 0.3 μM Arp2/3 complex and 13.5 μM profilin (circles). Further addition of 0.02 μM CP (diamonds) resulted in a distinct lag phase before polymerization was initiated. **e** The addition of 20 nM CP to 3 μM actin, 0.3 μM VCA, 0.3 μM Arp2/3 and 13.5 μM shifted actin polymerization to the membrane. Two different binding modes occurred. Confocal images of the equatorial plane of the vesicles show both modes: flat cortex (left side) and flat actin domains (right side). Scale bars are 20 μm

nM, the concavely bent domains were observable for several CP concentrations (from 40 to 120 nM) (Supplementary Figure 4).

Only at low (20 nM) and high (120 and 180 nM) CP concentrations closed cortex formation was observable (40% and 60%, respectively) (Fig. 2d and Supplementary Figure 4). However, cortex morphology was distinct: at 20 nM CP the cortex appeared as an almost homogenously grown network (Supplementary Figure 2), whereas at 120 nM CP the cortex consisted of many small actin domains, which overlapped with each other (Fig. 2d). In the latter case, concavely bent areas were observable within the cortex in 30% of the vesicles (Supplementary Figure 4). A further increase of CP to 180 nM led to the appearance of volume spanning networks (Fig. 2d and Supplementary Figure 4). Importantly, only one type of domain morphology was exclusively present in each vesicle; no vesicles were observed where two types of morphologies appeared simultaneously.

We also labelled VCA with a fluorescent dye to ensure that the observed domain formation was induced by actin polymerization and not by VCA clustering. The distribution of VCA in the absence of any actin polymerization was homogeneous at the membrane for several hours (Fig. 2e). Arp2/3 complex labelled in the same manner bound homogeneously to the membrane via the membrane adsorbed VCA molecules in the absence of any actin polymerization (Fig. 2f). In the presence of localized actin polymerization VCA started to concentrate at the polymerization sites (Fig. 2g). Thus, domain formation was not induced by any VCA clustering. Rather nucleation seeds of actin formed randomly at the membrane and led to an increased accumulation of VCA in the polymerizing actin domains.

**Domain nucleation and growth is controlled by CP.** To characterize the growth process of the differently shaped domains we obtained time-dependent snapshots at 40 and 120 nM CP (Fig. 3a, b). We observed that the time of actin seed formation at the membrane strongly depended on the CP concentration. At 40 nM CP nucleation was initiated rapidly as small actin patches

appeared immediately, whereas at 120 nM CP seeds were only recognizable at a later time (4 min). After 4 min domain formation was fully completed at 40 nM CP and after 7 min at 120 nM CP. The formation of concave actin domains was observed at both CP concentrations, whereas the formation of protrusions appeared only at 40 nM CP (Supplementary Figure 5). Protrusion formation was quick as compared to the formation of concave domains. Membrane deformations could be observed after 7 min (Fig. 3a, b), whereas a membrane protrusion became instantly visible (Supplementary Figure 5).

These observations indicate that the finite pool of actin monomers must compete with all nucleation sites such that the monomers are already consumed before the nucleation sites can grow large enough to fuse together. Consequently, separate actin domains of different filament density emerge at the membrane. In accordance, a correlation between the number of domains in a vesicle and the type of morphology was found (Supplementary Figure 6). The number of domains was lower for protrusions compared to concave domains and flat domains. As soon as the monomer pool was fully consumed the heterogeneities within the network remained stable for several hours.

For a better characterization of CP-dependent domain growth, we performed total internal reflection fluorescence (TIRF) measurements of actin domains growing on a His-VCA functionalized lipid monolayer at different CP concentrations. We observed that domain growth became more confined and the number of domains increased with an increase of CP (Supplementary Figure 7). This is consistent with the CP-dependent results from the actin-pyrene measurements. Here, the lag phase before polymerization initiation increased with an increasing CP concentration, whereas polymerization speed decreased with an increase of CP (Supplementary Figure 8).

**NMM II is most efficient to bend actin domains at high CP.** The diameter of the concavely bent domains was about 5 μm for 40 and 60 nM CP, respectively, while at 120 nM CP a significant

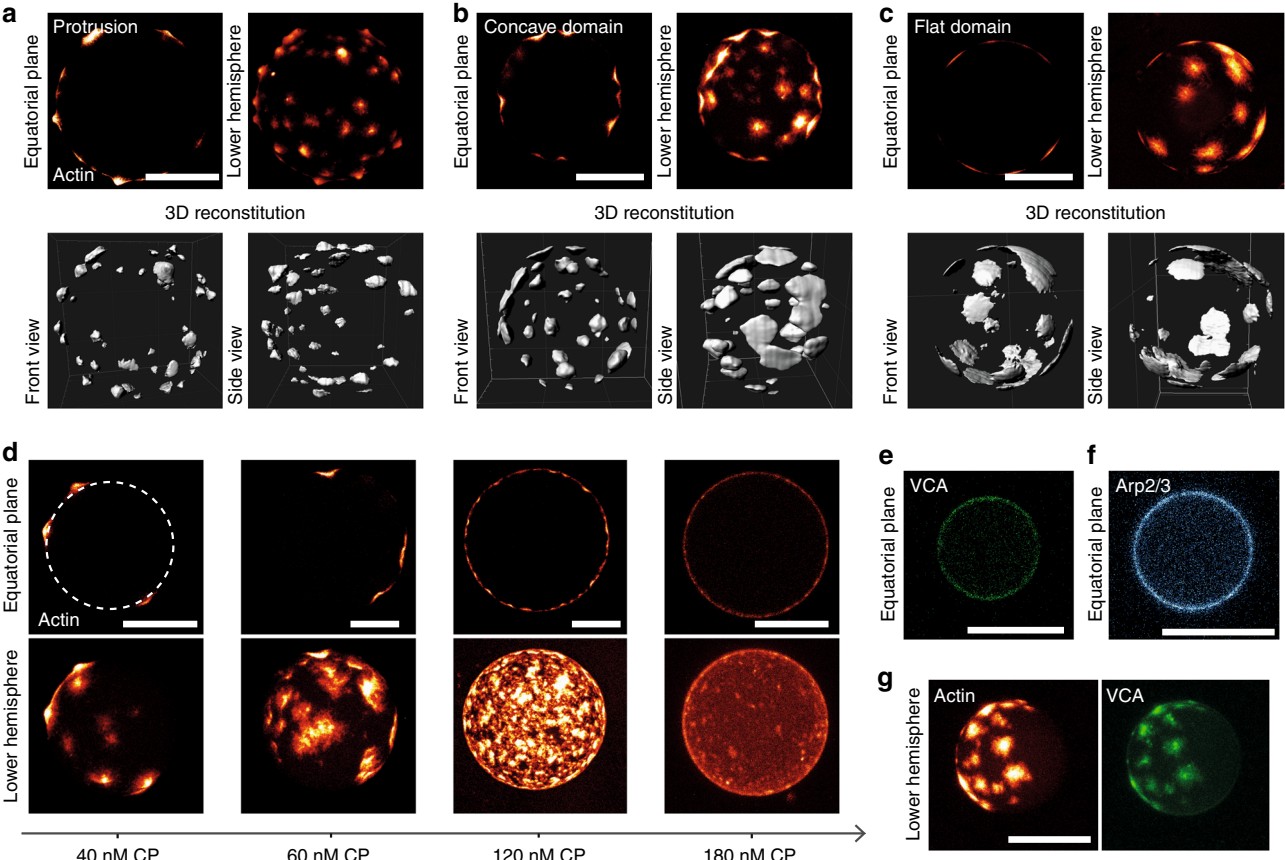

**Fig. 2** Distinct domain morphologies appear at the membrane dependent on CP concentration. **a-c** The equatorial plane of the vesicle, a confocal z-projection of the lower vesicle's hemisphere, the front and side view of digitally 3D reconstituted membrane localized actin domains are shown for the three different shape deformations: **a** Spatially confined actin protrusions emerged at the membrane, which pushed the membrane forward and such induced a positive bending of the membrane. **b** Thick and laterally expanded actin patches grew at the membrane. The membrane sided surface was concave and imitated negative curvatures within the membrane. **c** Thin and laterally expanded actin domains grew along the membrane and did not induce any shape deformations of the membrane. **d** The observed actin shapes depended on CP concentration. Shown are exemplary confocal micrographs of Ni-NTA containing vesicles at different CP concentrations: actin protrusions (40 nM CP), concave actin domains (60 nM CP), concave actin cortices (120 nM CP) and undeformed cortices (180 nM CP). Top panels show the equatorial plane and bottom panels a z-stack projection of the lower hemisphere of the vesicles. The white-dashed line indicates the vesicle's membrane. **e, f** Domain formation of actin was not induced by VCA or Arp2/3 complex clustering. **e** A total concentration of 0.3 μM encapsulated VCA bound homogeneously to the Ni-NTA membrane in the absence of actin and any ABPs. A 25% of VCA was labelled with Atto 647N. **f** A total concentration of 0.3 μM of Arp2/3 bound uniformly to the Ni-NTA vesicles in the presence of 0.3 μM unlabelled VCA. A 35% of the Arp2/3 was labelled with Atto 647N. **g** VCA clustering induced by actin polymerization is shown. The full system containing 3 μM actin, 0.3 μM VCA, 0.3 μM Arp2/3 complex, 60 nM CP and 13.5 μM profilin was encapsulated inside of Ni-NTA vesicles. A 25% of VCA was labelled with Atto 647N and 10% of actin was labelled with Atto 488. All scale bars are 20 μm

decrease of their size to 3.2 μm was observable (Fig. 4a and Supplementary Figure 9). This correlated inversely to the domain curvature, which increased with an increasing CP concentration from $-0.1\,\mu m^{-1}$ to $-0.13\,\mu m^{-1}$ (Fig. 4b). Scatter plots also confirmed that the domain size directly correlated to the domain curvature. Only at domain diameters lower than 4 μm, we measured curvatures higher than $-0.2\,\mu m^{-1}$ (Fig. 4c). Curvatures up to $-0.5\,\mu m^{-1}$ could be observed, but in none of the vesicles the formation of strong membrane invaginations nor any fission events were seen. The interplay of the polymerized actin gel with the membrane was not strong enough to induce full detachment of the invaginations. Thus, we added NMM II to the solution prior to encapsulation. The resulting contractile response of the motors within the actin network was strongly dependent on the CP concentration.

At 20 nM CP the presence of NMM II resulted in the formation of star-like actin clusters with a diameter of a few micrometres (Fig. 4d). In some cases, global deformations of the vesicle into non-spherical shapes were observable. Here the

NMM II induced tensions were balanced by the osmotic pressure[37].

The addition of NMM II at a CP concentration of 60 nM resulted in formation of domains, which remained connected to the membrane. The diameters of the domains contracted from 5 to 3.2 μm (Fig. 4a) and small membrane invaginations were discernible in 52% of the vesicles. Complete fission events occurred in 3.7% of the vesicles (Fig. 4d and Supplementary Figure 10). A slight curvature increase of the concave domains from $-0.1\,\mu m^{-1}$ to $-0.12\,\mu m^{-1}$ was observed (Fig. 4b).

A further increase of CP to 120 nM was sufficient to promote fission events of membrane invaginations in 27.5% of the vesicles (Fig. 4d and Supplementary Figure 10). This was consistent with a curvature increase from $-0.13\,\mu m^{-1}$ to $-0.22\,\mu m^{-1}$ (Fig. 4b). The domain diameters remained constant (Fig. 4a).

**Structure of actin domains depends on CP.** To understand how CP affects the network growth process we modelled the membrane localized network growth by a system of consecutive

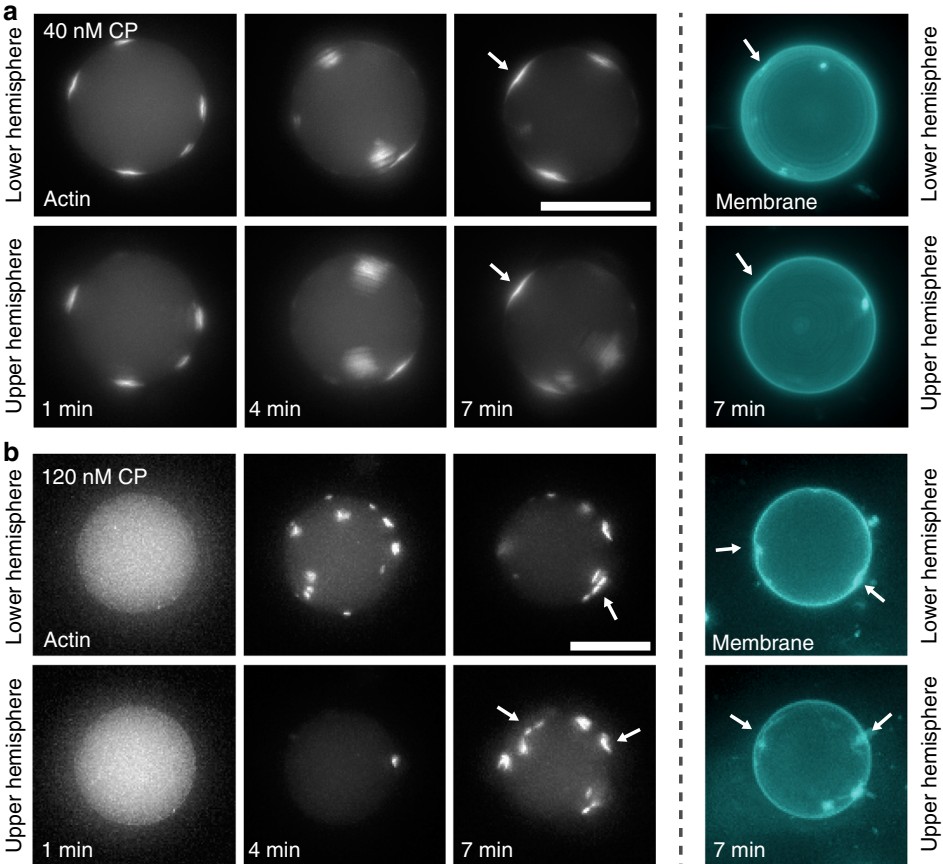

**Fig. 3** Snapshots of the temporal evolution of actin domain formation controlled by CP concentration are shown. **a**, **b** Projections of epifluorescence *z*-stacks show the temporal evolution of actin domain formation at the membrane in the presence of 40 and 120 nM CP, respectively. **a** Domain formation of actin at the membrane was fast in the presence of 40 nM CP. Already one minute after production has been finished, domains were visible at the membrane. Domain growth continued over time and a bending of the membrane was observable after 7 min (arrows). **b** Domain formation at 120 nM CP was slowed down. One minute after production domains were not observable, after 4 min domains became visible and after seven minutes a concave shaping of the membrane was induced (arrows). Please note that during the acquisition the vesicles were freely rotating and thus the domains occur at different positions in the images. To avoid bleaching effects only very limited number of snapshots was taken. All scale bars are 20 µm

elementary reaction equations[14,33]. All involved protein pools are finite, and equilibrate either by two-dimensional (2D) or three-dimensional (3D) diffusion processes. The network growth and chemical composition are determined by the two antagonistic key players Arp2/3 and CP. The Arp2/3 complex must be activated by VCA (Fig. 5a), which is localized at the membrane and needs to bind an actin monomer prior to activation. Upon activation, activated Arp2/3 complex (Arp2/3*) induces the nucleation of new filaments by branching from an existing filament ends. CP terminates barbed end elongation, which results in shorter filaments and less branching possibilities. The elongation and branching rates are set by the available monomer pool and the availability of Arp2/3*. As all monomeric actin is incorporated into the network at the membrane, we computed the network growth as the density of barbed ends with time (Fig. 5b). Hence the time dependence reflects the vertical network growth.

At the start of the simulation all VCA is bound to the membrane in the inactive state, and all other proteins are monomeric inactive and in the bulk. The VCA-G actin complex formation and subsequent activation of Arp2/3 is almost instantaneous, while the nucleation of first actin filaments are delayed. These filaments are necessary to initiate network growth (seeding) and once they are formed a rapid increase in the density of barbed ends is observed (initial growth). As the pool of available Arp2/3* is finite and capping antagonizes branching, the growth rates of barbed ends decrease (transient phase) and barbed end density finally reaches a steady state. The growth processes towards the steady state are controlled by CP (Fig. 5b).

To observe the steady state, we first discuss the case of a distant infinite reservoir of proteins, in which concentration gradients could still occur. The system started far away from equilibrium. This resulted in a branching dominated by the initial growth phase that transitioned into the CP concentration-dependent steady-state. Increasing CP concentration lowered the steady-state barbed end density (Fig. 5c and Supplementary Figure 11). The steady-state concentration of available Arp2/3* was higher with increasing CP, as the available barbed end density was decreased, which in turn resulted in a lower branching rate (Fig. 5d, e).

Directly after a seeding event of a domain only few barbed ends were present. Increasing CP concentration resulted in a decrease of the growth rate, as also experimentally observed in pyrene assays. The low barbed end density resulted in an accumulation of Arp2/3*, as VCA activated more Arp2/3 complex than requested by the network (Fig. 5c, d). Subsequently, the high Arp2/3* concentration was rapidly transformed into branches due to the self-amplifying reaction. The binding rate of CP is slower and thus it needed some time that CP could follow up and counteracted this increase of the activated Arp2/3* and barbed

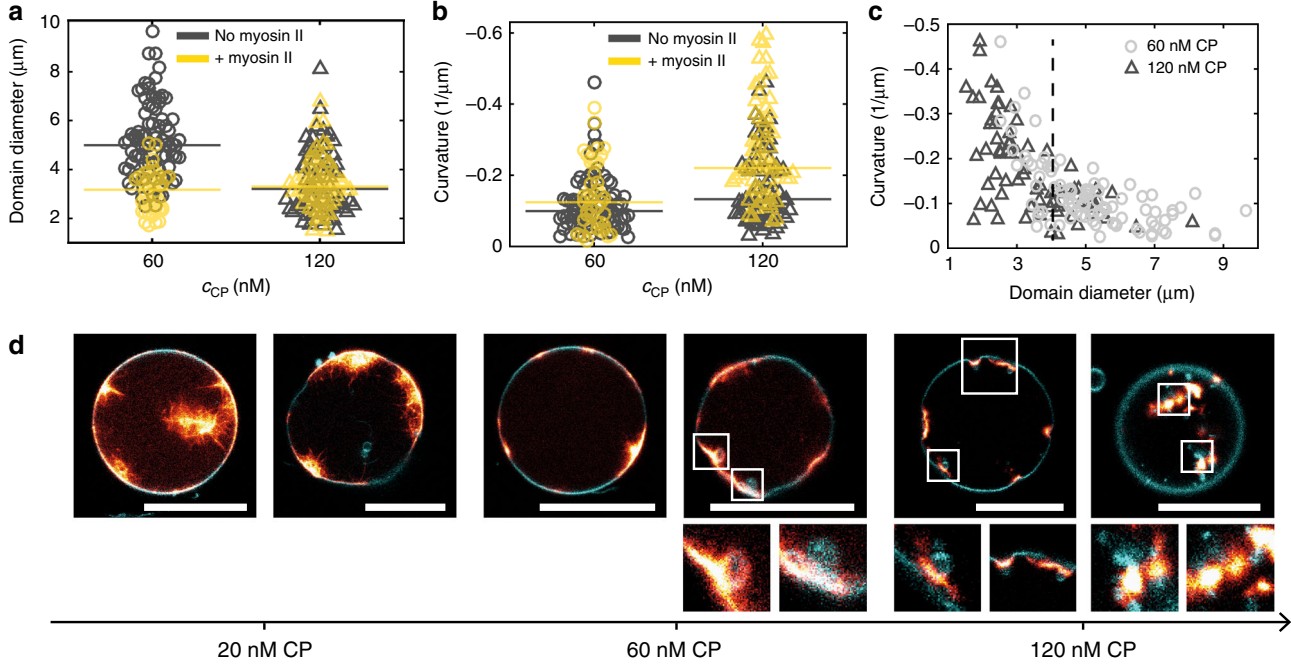

**Fig. 4** Contractile network behaviour depends on CP. **a**, **b** Domain diameter and curvature of negatively bent domains were measured in the absence or presence of non-muscle myosin II. **a** The median diameter decreased from 5 μm at 60 nM CP (grey circles) to 3.2 μm at 120 nM CP (grey triangles). Upon addition of 5 μM non-muscle myosin II to the solution prior to encapsulation the domain diameter contracted to 3.2 μm at 60 nM CP (yellow circles) and remained constant at 120 nM CP (yellow triangles). **b** The median curvature increased from 0.1 μm$^{-1}$ at 60 nM CP (grey) to 0.13 μm$^{-1}$ at 120 nM CP (grey). The addition of 5 μM non-muscle myosin II increased the curvature to 0.12 μm$^{-1}$ at 60 nM CP (yellow) and to 0.22 μm$^{-1}$ at 120 nM CP (yellow). **c** Scatter plots of the domain curvature vs. the domain length show that the domain diameter directly correlated to the domain curvature. Only small domains (<4 μm) had curvatures up to 0.5 μm$^{-1}$. **d** Confocal images of the equatorial vesicle's plane showed the interactions of the contractile actin network (red hot) with the membrane (cyan). At 20 nM CP the network detached from the membrane and was pulled towards the bulk of the vesicle. At 60 nM CP membrane invaginations (insets) emerged from the membrane. At 120 nM CP membrane invaginations (left inset) and fission events (right insets) could be observed. Scale bars are 20 μm

end density (Fig. 5e). Together with the steady-state conditions two distinct cases emerged: for low CP (40 nM), the increasing barbed end density slowly levelled off to a high steady-state concentration. Simultaneously, the Arp2/3* concentration rapidly reached its steady-state value. The balance between the rates of Arp2/3* activation and capping resulted in a transient and weak peak of branching events. For high CP (80, 120 nM), the initial increase in barbed end density was slower than at low CP, the resulting barbed end maximum was much higher than the steady-state level. This resulted in a peak of the barbed end densities.

Next, we considered systems with finite reservoirs and observed the spatial spreading along the membrane, which was implemented as a diffusive random walk of the barbed ends scaled by the local polymerization speed in a time dependent effective 2D simulation. In addition to the kinetic effects, now the depletion of the molecules during the growth process established gradients in the barbed end densities. The depletion of CP and actin monomers was found to be equivalent to continuously changing the starting conditions. We adjusted the simulation volume to take account for the observed number of domains that share the protein pools in the vesicles. At all CP concentrations, the behaviour discussed above was modulated by the global protein pool depletion. This corresponded to continuously dropping steady-state levels, which resulted in an initial peak for all cases.

At low CP concentration (40 nM), a continuous growth with a transient peak of barbed ends density was observable, which resulted from the drop of the steady-state levels. As the actin pools get used up, the maximum of the peak decreased continuously during spreading along the membrane. The growth

and spreading of the barbed end density along the membrane resulted in a flat radial distribution of barbed ends after 40 s (Fig. 5f). Raising the CP concentration (80 nM), we again observed the peak of barbed end densities due to the excess in Arp2/3*. As the network spread along the membrane, filaments entered membrane regions that also exhibited this excess in Arp2/3*. This resulted in a polymerization wave, creating a propagating high barbed end density at the rim of the domain and a lower density in the middle (Fig. 5f).

At the highest here studied CP concentration (120 nM), the concentration profiles became flat again (Fig. 5f). This was due to the implementation of the number of domains in the simulation, effectively reducing the available monomer pool. This additional depletion by competition of multiple domains shows that global conditions are also essential for the growth process of the domains.

Here, the experimentally observed nucleation at multiple sites due to the increased CP results in local variation of polymerization conditions. Overall, we observed CP-dependent radial patterns of the barbed end density that we relate to the geometry of the observed actin domains in vesicles, considering that network growth velocities directly depend on the barbed end density[17].

**CP overexpression induces vesicle formation in B16-F1 cells.** Finally, we tested the capability of CP to induce membrane deformations in cells. To this end we overexpressed heterodimeric CP in the widely used B16-F1 mouse melanoma cells using a doxycycline-inducible expression system. Quantification of the

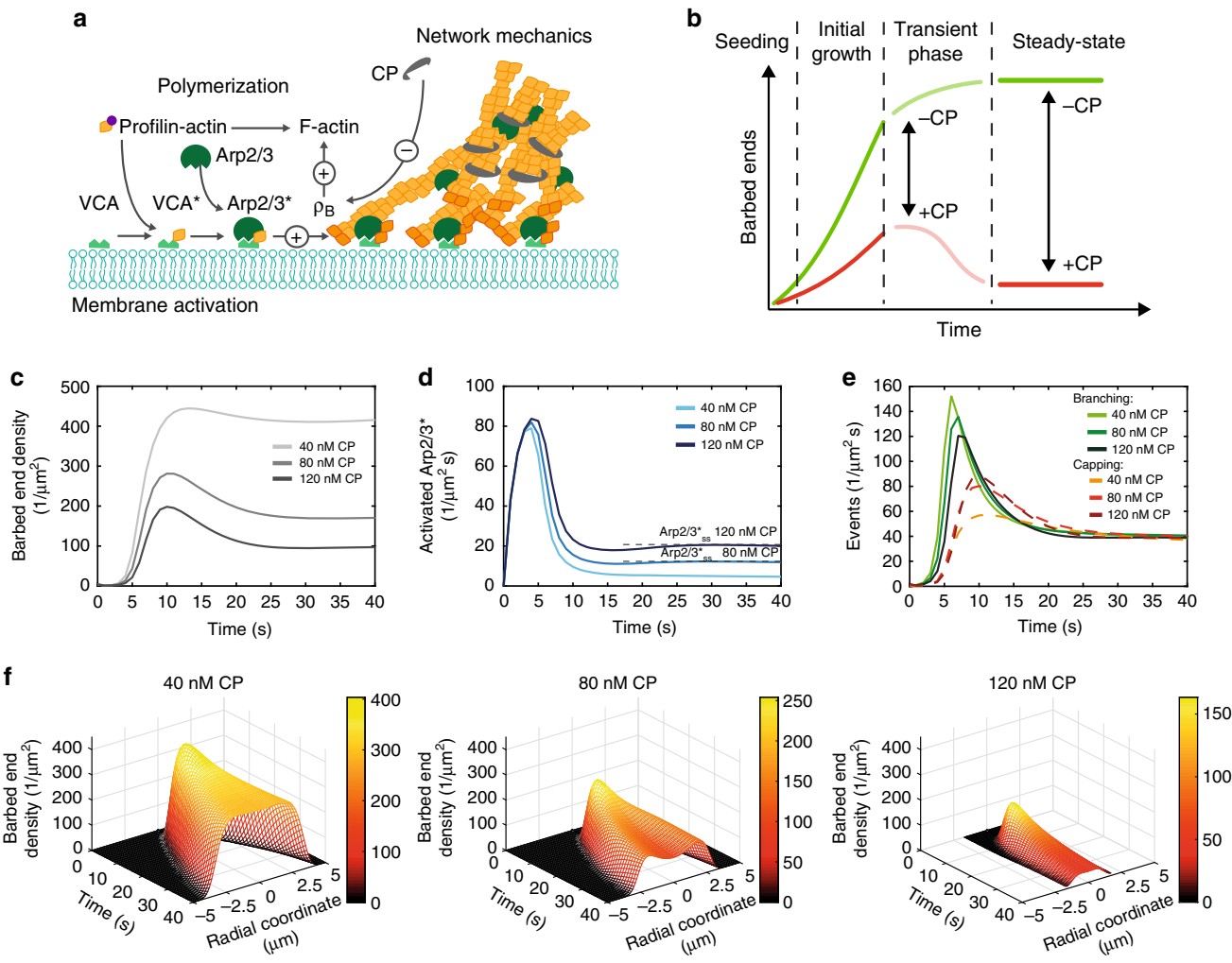

**Fig. 5** Localized actin polymerization is modelled in the presence of CP. **a** Schematic of membrane localized actin assembly kinetics: a finite reservoir of membrane activated Arp2/3 complex nucleates a branched network at the membrane. After polymerization start the available Arp2/3 complex reservoir must equilibrate towards a new steady-state concentration as the regeneration rate is slow. The balance between branching by Arp2/3 complex and capping by CP determines the final network mechanics leading two the observed shape deformations of the membrane. **b** Network growth can be observed by modelling the temporal evolution of barbed ends. Four different growth regimes occur. The seeding phase, where first barbed ends are formed, is followed by an initial rapid growth. Finally the barbed ends level off to a steady-state by passing a transient phase. The growth processes depend on the concentration of CP. **c–e** The temporal evolution of barbed ends (**c**), activated Arp2/3 complex (Arp2/3*) (**d**), branching and capping events (**e**) at infinite protein pools is shown. The final steady-state of barbed end density and Arp2/3* was set by CP concentration. The balance between branching and capping events controlled the equilibration process of network growth towards the steady state. **f** The spatio-temporal network growth was modelled along the membrane at finite protein pools. Two distinct barbed end patterns emerged. At 40 and 120 nM CP a flat barbed end distribution was observed and at 80 nM a double-peaked density distribution occurred

CP beta 2 chain by densitometry revealed a 2.1-fold over-expression on average (Supplementary Figure 12), but given the high variability of the expression levels in the cells as assessed by immunofluorescence intensity of the HA-labelled CP alpha 1 chain, we assume that CP overexpression in the strong over-expressors must be considerably higher. Ectopically expressed CP accumulated at the leading edge, and consistent with previous work reporting on the explosive increase of filopodia in CP-depleted B16-F1 cells[29], we found that the formation of micro-spikes was largely inhibited in the strong CP overexpressors, as opposed to the numerous microspikes in the lamellipodia of untransfected control cells (Fig. 6a, b). As expected the thickness of the lamellipodium also decreased from 1.7 to 0.7 μm in cells overexpressing CP (Fig. 6c). Most importantly, however, the cells strongly overexpressing CP exhibited a substantial increase of actin-containing vesicles by about 50% (Fig. 6d). However, since

neither the endosomal marker Rab7, nor the lysosomal marker LAMP1 or the fluid-phase marker TRITC dextran was found to colocalize with the majority of these CP-containing vesicles, we could exclude the possibility that the formed vesicles were regular endosomes, lysosomes or vesicular structures entrapped by micropinocytosis (Supplementary Figure 13). Our observations therefore clearly corroborate the view that an increase of CP concentration is already sufficient to induce and shift membrane deformations from concave to convex in vivo, but further implies that overexpression of CP alone without upregulation of the other components of the machinery and appropriate signalling is not sufficient to increase the number of endocytic vesicles.

## Discussion

Our work shows that already a variation of CP concentration between 20 and 180 nM results in distinct growth conditions

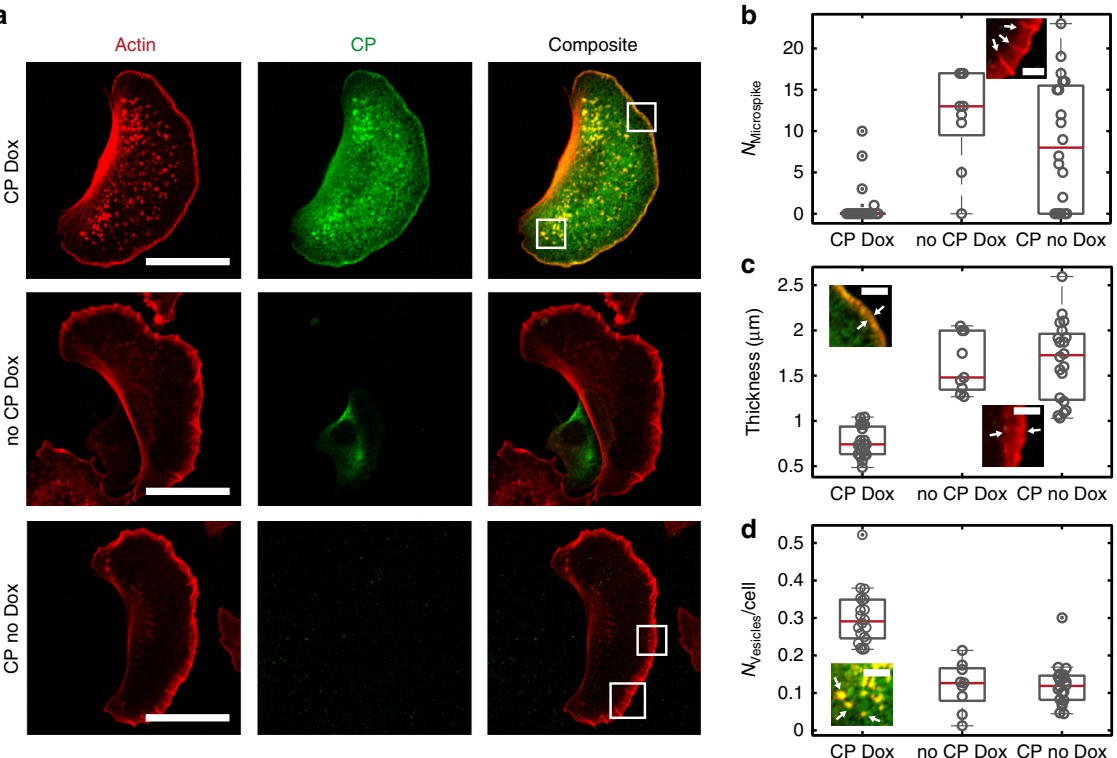

**Fig. 6** CP affects membrane deformations in vivo. **a** The distribution of phalloidin stained F-actin (red) and of CP visualized by HA-Tag antibody labelling (green) in fixed cells is shown: In the top cells overexpressed CP after induction with 1 µg/ml Doxycyclin (CP Dox, $N = 21$). In the middle cells were not overexpressing CP but treated with 1 µg/ml Doxycyclin (no CP Dox, $N = 10$) and in bottom cells contained the expression plasmids but were not treated with Doxycyclin (CP no Dox, $N = 21$). **b** CP overexpression resulted in the complete loss of microspikes. **c** CP overexpression decreased lamellipodium thickness. **d** Formation of actin-containing vesicle was increased in CP overexpressing cells. The number of vesicles was normalized to cell area. Arrows indicated the evaluated objects. The central red line in the boxplots indicates the median. The bottom and top edges of the box mark the 25th and 75th percentiles, respectively. The whiskers got to the most extreme considered data point. Not considered data points are marked by a dot in the centre of the data point. Scale bars in **a** are 20 µm and in **b–d** are 3 µm

within networks formed by Arp2/3 complex, which induce either positively or negatively curved membrane deformations. Low CP concentrations ensure fast network nucleation and growth at the membrane, which results in the formation of homogeneous cortices. Higher CP concentrations slow down nucleation efficiency of the network such that a few nucleation sites appear at the membrane and the finite actin pool is partitioned between those. Distinct actin domains form at the membrane, which ultimately induce shape deformations in the membrane. High CP concentrations slow down network extension, thus over time slow and confined growing domains emerge at the membrane until they are densely packed to form closed cortices. This balance between nucleation and growth speed finally leads to the observed appearance of volume spanning networks at the highest CP concentrations studied here (180 nM). The temporal resolved TIRF-measurements of domain growth on a lipid monolayer dependent on the CP concentration confirm that Arp2/3 induced nucleation and domain growth correlate with the CP concentration. Confocal measurement of the distribution of fluorescent VCA at the vesicle's membrane identified an up-concentration of the Arp2/3 activator during polymerization, which is in agreement with experimental and theoretical work[1,10].

By kinetic simulations we identified that the out of equilibrium nature of the reaction kinetics of actin and its binding proteins are responsible for the observed deformations. The coupled kinetic equations result in two distinct non-uniform polymerization patterns leading to inhomogeneous barbed end density profiles. These can be directly linked to different protrusion

velocities of the growing networks based on a ratchet model for membrane protrusions[39]. Compared to in vivo and most in vitro studies (that use a ~100-fold VCA density[14,15]), the number of barbed ends in our model system is low ($<=450/\mu m^2$) and therefore we assume that protrusion velocities are not dominated by monomer depletion, but rather by membrane tension[17,40]. In this regime the load per filament decreases with increasing network density. Consequently, the protrusion velocity increases exponentially with an increasing number of barbed ends[17]. For low CP concentrations the number of barbed ends is constant along the membrane. Therefore, the network grows homogeneously and an actin protrusion is formed. For higher CP concentrations the number of barbed ends is lower in the domain centre than at the domain edges. Thus, the network protrusion is faster at the domain edges and slower in the domain centre. A concave actin domain is formed. Although the kinetic simulations are able to rationalize the emerging structures, it remains a major challenge to combine the kinetic simulations with mechanical responses of the complete system.

Our experiments further reveal that CP is an important regulator of the network structures in cells. While upregulation of CP leads to an increased formation of actin-containing vesicles, decrease of lamellipodium width and the inhibition filopodia formation, a down regulation was previously shown to result in an explosive increase of filopodia and complete suppression of lamellipodia formation[17]. The strong interdependence of the network structures results from a homeostasis effect of the actin cytoskeleton structures in cells, as first demonstrated to be an

important mechanism in fission yeast[41,42]. In this case, the limited actin monomer pool is split between the different actin structures, depending on the activation of the nucleation factors Arp2/3 complex and formin.

In our experiments the growth-induced network stresses alone were not sufficient to induce fission of the invaginations from the membrane in the in vitro system. By the addition of NMM II membrane fission events are induced in a CP-dependent manner and endocytic-type vesicles were formed in the in vitro system. The requirement of myosins is in accordance with observations in vivo[43–46]. Further studies are needed to shed light on the dynamics and physical mechanisms behind membrane fission events.

To conclude, we demonstrated that the growth conditions of branched actin networks are decisive for fulfilling their multiple tasks. CP controls in a concentration-dependent manner of the microscopic assembly kinetics, which in turn alters the mechanical properties of the growing network sufficiently to induce either negative or positive curvatures to the actin network and the attached vesicle membrane. A detailed finite-element analysis of the CP-dependent growth patterns is required to fully link microscopic actin assembly to mesoscopic membrane deformations. In cells the CP regulatory protein CARMIL and V-1 were identified to tightly control the amount of available CP[47–49], which could be decisive for cells to switch between the diverging demands of function. It is the balance between the presence of nucleation factors and CPs, which sets the final function of cytoskeletal structures via the assembly kinetics. This underlines the urgent need to understand the full reaction kinetics in networks beyond the steady-state conditions.

## Methods

**Protein purification.** G-actin was purified from rabbit skeletal muscle by a modified protocol of Spudich et al.[50]. It was stored at 4 °C in G-Buffer (2 mM Tris, 0.2 mM ATP, 0.2 mM CaCl$_2$, 0.2 mM DTT and 0.005% NaN$_3$, pH 8.0). G-actin was labelled with Atto 488 NHS-ester (Jena Bioscience)[51] at cysteine 374 or with N-(1-pyrenyl)iodoacetamide to perform pyrene assays[52]. Arp2/3 complex was extracted from pig brain and purified as described previously[53]. It was fluorescently labelled by adding a 3-fold excess of maleimide Atto 647N dye (Atto-Tec) and incubated while rotating for 2 h at −4 °C. Finally, the reaction was stopped by the addition of 1 mM DTT and the excess dye was removed by a NAP size-exclusion chromatography column (NAP-25 Columns, GE Healthcare). The C-terminal domain (VCA) of murine N-WASP was expressed in BL21 Codon Plus DE3-RIPL (Agilent Technologies) as a His-tagged protein overnight in LB medium. After purification by Ni-NTA affinity chromatography in SoniBuffer (50 mM Na$_2$HPO$_4$, 50 mM NaH$_2$PO$_4$·2H$_2$O, 300 mM NaCl, pH 8.0), the protein was dialysed against storage buffer (20 mM Tris/HCl, 100 mM KCl, 1 mM MgCl2, 5 mM EGTA, 2 mM DTT, pH 7.0) and stored at −80 °C. We kept the histidine-tag to enable the binding of VCA to Ni-NTA lipids. VCA was labelled by the addition of a 3-fold excess of maleimide dye Atto 647N (Atto-Tec) and incubated for 2 h at 4 °C. Afterwards it was quenched by 1 mM DTT and the excess dye was removed by size-exclusion chromatography using a NAP-25 column (GE Healthcare). Profilin (mPfn2a) was expressed in BL21 Codon Plus DE3-RIPL (Agilent Technologies) as a glutathione S-transferase (GST) fusion protein. After its purification it was dialysed against 20 mM Tris (pH 7), 150 mM NaCl, 1 mM DTT and stored at 4 °C[54]. The GST-tag was removed by proteolytic cleavage after purification. The mouse α1- and β2-subunits of CP[55] cloned into pRSFDuet™-1 (Novagen) were expressed in BL21 Codon Plus DE3-RIPL (Agilent Technologies). The cells were grown at 37 °C until the $A_{600\,nm}$ of 0.6 was reached. The expression was induced by the addition of 0.5 mM IPTG and shaking was continued overnight. The cells were collected and resuspended in 20 mM Tris (pH 8.0), 250 mM NaCl, 1 mM EDTA, 5% glycerol. After centrifugation (30,000 r.p.m., 10 min, 4 °C) supernatant was mixed with 100 μl ANTI-FLAG M2 Affinity Gel (Sigma Aldrich) per 100 ml supernatant, rotated for 90 min at 4 °C and then centrifuged at low speed (500 r.p.m.). Beads were washed several times with the same buffer before adding 100 μl FLAG Peptide (Sigma Aldrich F3290-25mg lyophilized powder) per 1 ml buffer. After a 1 h incubation on a rotating wheel at 4 °C they were centrifuged for 2 min at 14,000 r.p.m. Supernatant with CP was dialysed against 10 mM Tris/HCl (pH 8.0), 50 mM KCl, 1 mM DTT overnight then flash-frozen in liquid nitrogen and stored at −80 °C. NMM II was purified from human blood platelets and stored at −20 °C in 60% sucrose following a standard protocol[56].

**Encapsulation.** The encapsulated protein solutions were mixed on ice directly before vesicle production. Here 3 μM G-actin, 300 nM VCA, 300 nM Arp2/3 complex, 13.5 μM profilin and the desired amount of CP were added to the polymerization buffer (10 mM imidazole, 3 mM MgCl$_2$, 30 mM KCl, 1 mM EGTA, pH 7.4). We replaced 10% of unlabelled G-actin by covalently labelled Atto 488 actin to mark the network structures fluorescently. To measure the contractile behaviour dependent on CP, 5 μM non-muscle myosin II were added additionally to the encapsulation mix. In all experiments the amount of all protein buffers was kept constant. Only 300 nM VCA (25% were labelled with Atto 647N) were encapsulated in the absence of any other proteins to study effects of VCA clustering on domain formation. To study VCA induced Arp2/3 clustering, we encapsulated 300 nM of Arp2/3 (35% were labelled with Atto 647N) additional to 300 nM VCA. The osmotic pressure of the outside solution, which was prepared of 1 M glucose, was adapted 10 to 20 mosmol higher compared to the encapsulated solution. A total concentration of 0.5 mM lipids were dissolved in a mixture of 14% mineral oil (Sigma Aldrich, M3516), 80 % silicone oil (Roth, 4020.1) and 6% decane (Sigma Adlrich, D901) as previously described[37]. The final lipid oil mixture contained 87.4% Egg-PC (Sigma Aldrich, P3556), which was dissolved at 50 mg/ml in a chloroform/methanol mixture (9:1, v/v), 10% Ni-NTA (Avanti Lipids, 790404 C) and 2.5% PEG2000 PE (Avanti, Lipids, 880160 C). To label the membrane 0.06% of the fluorescent lipid Texas Red (Thermo Fischer, T1395MP) was added to the lipid mix. Vesicles were produced for 5 min at 5 °C using a slightly modified protocol of the recently published cDICE encapsulation method[38]. We used capillaries with a diameter of 40 μm to prevent protein clogging and adapted the speed of rotation to 1200 r.p.m. The capillary was injected in a solution containing 70% decane and 30% mineral oil. The vesicles were imaged 30 min after production and remained stable for several hours. All reported experiments we performed at least twice with different preparations of proteins. Quality of protein charges were determined by bulk assays.

**Pyrene assays.** The polymerization kinetics of the encapsulated protein mixtures were measured by pyrene assays. To this end 20% of the unlabelled G-actin were replaced by pyrenyl actin. The measurements were carried out with the microplate reader SpectraMax M5 (Molecular Devices).

**Imaging and data acquisition.** Bright-field and epifluorescence images of the vesicles were taken by a ×63 numerical aperture (NA) 1.3 oil immersion objective on a commercially built Leica Microscope DMI3000 B in combination with a Hamamatsu ORCA-ER camera. Confocal images were taken by a ×63 NA 1.4 oil immersion objective with on a Leica TSC SP5. The temporal evolution was observed by acquiring 3D stacks of epifluorescence snapshots with a ×100 NA 1.4 oil immersion objective or a ×40 NA (1.25–0.75) oil immersion objective on a commercially built Leica Microscope DMI3000 B. The 3D image acquisition was started one minute after finishing the encapsulation process. We acquired snap-shots of the 3D epifluorescence stacks of the vesicles every three minutes in order to avoid any artefacts by phototoxicity.

Imaging of fixed cells was performed with an LSM510Meta confocal microscope (Zeiss) equipped with a ×63/1.3 Plan-Neofluar objective using the 488 nm and 543 nm laser lines.

Confocal and fluorescent images were used to measure the domain length and curvature by ImageJ. Actin domains at the membrane were counted with the 3D object counter provided by ImageJ.

**Vector construction.** For the expression of human heterodimeric CP the bicistronic IRES Tet-On 3G inducible expression system (Clontech) was employed. pTRE3G-IRES is a Tet-inducible, bicistronic mammalian expression vector designed to coexpress two genes of interest under the control of the Tet-responsive promoter pTRE3. pCMV-Tet3G expresses Tet-On 3G, a tetracycline-controlled transactivator that exhibits high activity in the presence of the inducer doxycycline. To generate pTRE3G-IRES-HA-CPα1, the corresponding sequence was amplified by PCR from cDNA as a MluI-BamHI fragment using the primers 5′-CGCACGCGTATGTATCCTTACGACGTGCCTGACTACGCCCATGGCC-GACTTCGATGATCGTGTG-3′ and 5′-GAGGGATCCTTAAGCATTCTG-CATTTCTTTGCC-3′ and inserted into the same sites of the second multiple cloning site (MCS2) of pTRE3G-IRES. The longer upstream primer contained additional sequences to allow expression of the α1 subunit with an N-terminal HA epitope (aa YPYDVPDYA) for later immunostaining. The corresponding sequence encoding the β2 subunit was amplified by PCR as a SalI-EagI fragment using the primers 5′-CGCGTCGACATGAGTGATCAGCAGCTGGAC-3′ and 5′-GAGCGGCCGTTAGCATTGCTGCTTTTCTCTTC-3′ and inserted into the same sites of MCS1 in pTRE3G-IRES-HA-CPα1 yielding pTRE3G-IRES-HA-CPα1/CPβ2. All sequences were verified by DNA sequencing.

**Cell culture and transfections.** B16-F1 mouse melanoma cells (ATCC CRL-6323) were grown in DMEM with 4.5 g/l glucose (Lonza) supplemented with 10% FCS (Biowest), 2 mM UltraGlutamine (Lonza) and 1% of a 100× penicillin/strepto-mycin stock solution (Biowest) at 37 °C and 5% CO$_2$. Absence of mycoplasma was routinely checked by the VenorGeM Mycoplasma Detection Kit (Sigma). Transfections were carried out with JetPrime (Polyplus), according to the manufacturers'

protocols at a 4:1 ratio of pTRE3G-IRES-HA-CPα1/CPß2 and pCMV-Tet3G. Prior to transfection the cells were seeded on glass coverslips coated with 25 μg/ml laminin (Sigma) overnight. Induction of CP overexpression was induced by addition of 1 μg/ml of Doxycyclin (Clontech) 4 h after the transfection.

**Fixation, labelling and analyses of cells**. After 24 h of induction, the cells were fixed with 4% paraformaldehyde in PBS for 20 min, extracted with 0.1% Triton X-100 in PBS for 1 min and stained with fluorescent phalloidin to visualize filamentous actin and with anti-HA monoclonal antibody 12CA5 (undiluted hybridoma supernatant), which is a subclone of H26DO[57], to visualize the HA epitope. Atto550-phalloidin (1:200 dilution, # AD 550-81; Atto-Tec) was used to stain F-actin and secondary goat-anti-mouse Alexa488-labelled antibodies (1:1,000 dilution; #A32723) were from Thermo Fisher. Anti-Rab7 antibody (1:200 dilution; #9367) was from Cell Signalling, anti-LAMP1 antibody (1:200; #ab24170) was from Abcam, and lysine-fixable TRITC dextrane (1 mg/ml; 70,000 MW, #D1818) was from Thermo Fisher. For immunohistochemistry, secondary Alexa-555-conjugated goat-anti-rabbit polyclonal antibodies (1:1,000 dilution; #A21429; Invitrogen) were used. Immunoblotting was performed according to standard protocols using undiluted hybridoma supernatant of anti-HA monoclonal antibody 12CA5, anti-CP-beta monoclonal antibody (1:200; #sc-136502; Santa Cruz Biotechnology), and anti-GAPDH monoclonal antibody (1:1000; #6C5; Calbiochem). Primary antibodies in immunoblots were visualized with phosphatase-coupled anti-mouse IgG (1:1,000; #115-055-62; Dianova). Lamellipodium thickness was determined by ImageJ. The phalloidin fluorescence intensity in lamellipodia was measured at four different positions to calculate the full width half maximum for two independent experiments ($n = 2$). For counting the number of CP-containing vesicles the Matlab-based program SpatTrack[58] for denoising was used. Subsequently the data was segmented by ImageJ.

**Code availability**. The COMSOL code for the simulations is available from A.R.B. upon reasonable request.

**Data availability**. The data that support the findings of this study are available from A.R.B. upon reasonable request.

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

## Acknowledgements

Research was supported by the Deutsche Forschungsgemeinschaft via the SFB863 and FA330/11-1 (J.F.) and by the ERC via the project SelfOrg. The continuous support by the Nanosystems Initiative Munich is acknowledged. A.R.B. acknowledges the hospitality of the Miller Institute for Basic Research in Science at the University of Berkeley.

## Author contributions

K.D., F.C.K, J.F. and A.R.B. planned the experiment, K.D. and P.B. carried out the experiments; J.F carried out the CP overexpression experiments in the B16-F1 cells. F.C. K. and F.B. performed the theoretical modelling; K.D. and A.R.B. performed data analysis; C.J.C. provided important insights into the elastic modelling of actin gels; K.D., F.C. K., J.F. and A.R.B. wrote the paper. All authors reviewed and revised the manuscript.

## Additional information

**Competing interests:** The authors declare no competing interests.

