## [Peer Review File · Nature Communications]

Reviewers' comments:

Reviewer #1, an expert in actin assembly at interfaces, has submitted his/her report as a PDF

Manuscript#: NCOMMS-17-09355-T

Capping protein controlled actin polymerisation shapes lipid membranes

By K. Henneberg, F. C. Keber, P. Bleicher, C. J. Cyron, J. Faix and A. R. Bausch

In the paper by Henneberg et al. the authors combine *in vitro* studies with computer simulation to study shape deformations in membrane induced by arp23 branched actin networks. The authors suggest that capping proteins play a major role in this process by controlling the density of pushing filaments and their lateral distribution along the membrane, thereby causing outward or inward deformation at low and high CPs concentrations, respectively. The authors also show the impact of myosin II on these deformation processes. The addition of myosin II enhances inward deformations and promotes endocytic events. The authors also perform *in vivo* experiments showing that CPs level affects the amount of actin found in endocytic vesicles and lamellipodia in cells.

Understanding the role of arp branched network density and structure on membrane shape deformation is an emerging subject and indeed timely. Nevertheless, the paper does not go deep enough, not in term of the experimental work nor in terms of the proposed model. The authors provide experimental and simulation results but do not provide explanations for their findings. Moreover, the authors do not provide any spatio-temporal data which is essential for understanding the impact of actin cytoskeleton structure on membrane shape deformation mechanism. Rather, the authors provide only results at the final system stage, with no information of how these systems involved into their final steady states. This is also true for the experimental part and for the simulations. Thus, despite the fact that the subject itself is important, this work does not provide deep insights and novel understanding on the role of actin cytoskeleton architecture on membrane deformations mechanism. Thus, in its current form, the paper does not meet the requirements for publication in Nat. Comm.

Here are my remarks:

1) Regarding the experimental work:

The authors do not provide any movie showing the spatio-temporal information on the dynamics of VCA localization, actin nucleation, network growth, formation of actin patches, and the emergence of inward and outward deformations. What are the time scales for each process? Why this information is not given? It is essential to understand the dynamics of these processes.

In their *in vitro* system the authors stress that the density of VCA at the membrane surface is 1200 molecules per microns², which gives a distance of ~30 nm between VCA molecules. This is a very high surface density that I would expect to result in a homogenous cortical network, regardless of CP density.

Since this is not the case, and the authors do observe actin patches, it is essential to show that VCA distribution on the membrane surface is initially homogenous.

If VCA distribution is indeed initially homogenous, this means that VCA clusterization occurs with time, which should greatly increase VCA concentration, and thus actin nucleation and network growth at these sites.

What about actin and VCA clusterization? What drives this process and what are the typical timescales of this process?. How CP concentration and myosin II impact this process? Finally, do actin polymerization impacts VCA clusterization (and/or vis-versa) – Note that previous experimental and theoretical works showed that there is a feedback mechanism between actin nucleators clusterization, local surface curvature, and actin polymerization (see for instance works by Basserau (expt), Gov (theory) and others). The authors should cite these works and discuss their relevance to this work.

Similar to the impact of CPs, the effect of myosin II motors is given but not discussed. It is unclear if and how the motors affect the dynamics of network assembly and actin clusterization. Myosin motors are also suggested to be responsible for membrane invagination and fission. What are the timescale of all these processes in comparison to systems that do not include myosin motors?. Where the motors localize? Labeling the motors would greatly facilitate the analysis of the data by determining their localization at the various steps of the process.

2) Regarding the model.

The authors use sets of equation rates to study the impact of CP on free (pushing) actin filaments barbed ends distribution at the membrane surface. In figure 4 the authors show the results of their simulations. The authors find that at low CP the distribution of filaments is higher at the center of the patch, which leads outward protrusion, while at a high CP concentration, the density of pushing filaments is higher at the edges, which lead to inward deformation. The authors do not explain the origin of this phenomenon and do not provide clear explanation to their results. In fact, it is unclear how CPs concentration controls this process and how it leads to these distinct distribution profiles.

In order to better understand the role of CP on membrane deformation, temporal evolution of the profiles of pushing barbed ends is required. In addition, spatio-temporal evolution of V , V^* , B , and A with time should be also given. How the size r of the system chosen affect the results of the simulation.

Since the authors observe actin patches formation, including VCA diffusion/transport into their equation seems to be essential.

Finally, while experimental work on myosin II is shown it is not addressed theoretically and should be added.

Other remarks:

- There are some typos mistakes along the text which render the reading difficult.
- Citation: here I bring few examples of errors or missing citations:

For instance, ref 6-8 are aimed to cite papers that discuss propulsion of spherical objects.

Ref 6 – is a theoretical paper, ref 7 does not show motion at all, and ref 8 discusses reconstitution of listeria propulsion (not spherical). Thus ref 6,7 should be replaced with correct relevant papers.

Ref 17-18 discuss lamellipodia to filopodia transition in vivo. The authors should also cite in vitro works.

Regarding the model (in addition to ref 1,2, the authors should also cite Carlsson AE who published numerous papers on the subject).

Reviewer #2, an expert in actin self-assembly (Remarks to the Author):

In this study, the authors revealed the role of CP together with the Arp2/3 complex in generating membrane deformation in a reconstituted system in vitro. Based on their observations, they challenged their model built in vitro, by manipulating the concentration of capping proteins in a cellular context.

Overall, I think that the formation of membrane protrusions at low CP concentration and negative membrane curvatures at high CP concentration is very interesting.

However, I found that the model (Figure 4) that explains why this is happening is not really clear. The authors should better explain why local actin assembly generated by high CP concentration generates invagination whereas more homogeneous actin assembly at low CP concentration generates protrusion. I was expected from these authors a detail model based on both kinetic of actin assembly but also on the mechanic aspect of the growing network and on membrane tension to challenge their main hypothesis.

In their cellular overexpression experiments, the authors will not only affect assembly by the Arp2/3 complex but also inhibits assembly by formins. Maybe the authors can also inhibit locally CP activity using carmil to see if decreasing CP concentration generates protrusion in their system.

I also predict that in the model increasing Arp2/3 concentration to a very high level should have a similar effect that high CP concentration. The authors should maybe generate a phase diagram varying CP and arp2/3 complex concentration for a defined actin concentration and determine how membrane deformation is affected.

Reviewer #3, an expert in actin regulation including capping protein, reviewed this work together with a colleague with expertise in modelling. Rather than collate their comments into one report, he chose to append them separately:

Reviewer #1

This manuscript treats the effects of Arp2/3-induced actin polymerization inside vesicles on the shape of the membrane. It is found that increasing the concentration of capping protein shifts the membrane morphology from having regions with outward curvature to regions having inward curvature. The addition of non-muscle myosin II drives the membrane bending farther, so that vesiculation is achieved in some cases. It is proposed that this transition results from polymerization overshoots induced by increasing levels of capping, which cause a characteristic spatial profile of barbed-end density, as well as force, along the membrane. The relevance to biological cells is demonstrated via capping-protein overexpression experiments on fibroblasts, in which enhanced endocytosis is observed.

The effect of capping protein on membrane morphology demonstrated here significant and provocative, especially in view of the experiments on cells. This effect to be relevant to a broad range of actin-induced membrane-bending phenomena. However, there are problems with the mathematical modeling:

i) The force plots in Fig 4c suggest a large net force acting on the actin network, on the order of $(200\text{pN}/\mu\text{m}) \times (3\ \mu\text{m})^2 = 600\ \text{pN}$. Such a large force would lead to very rapid motion of the actin network. As pointed out previously (Hassing, Julian E., et al. "Design principles for robust vesiculation in clathrin-mediated endocytosis," *Proceedings of the National Academy of Sciences* 114.7 (2017): E1118-E1127), the distribution of forces involves a balance of pulling and pushing forces so that the total force exerted by the actin network is close to zero. The force model should be revised to reflect this force balance.

ii) It is also not clear that the increased density of free barbed ends near the edges of the actin patch, obtained at high CP concentrations, will lead to pulling forces near the center. The proposed mechanism requires more rapid polymerization at the edges, so that the central region will be dragged back by retrograde flow. But in the absence of force, the increased density of barbed ends at the edges would cause monomer depletion (Khamviwath, Varunyu, Jifeng Hu, and Hans G. Othmer, "A continuum model of actin waves in *Dictyostelium discoideum*." *PloS one* 8.5 (2013): e64272), causing the network to grow more slowly at the edges than in the center. The inclusion of force would enhance this effect, since the pushing forces at the edges would slow polymerization, while the pulling forces in the center would speed polymerization. So at first glance the "overshoot" mechanism seems to give the wrong sign for the membrane bending.

There are also some minor points/suggestions:

ii) The authors use the terminology "spikes" for the actin morphology at low CP concentrations, but I do not see spikes. Is there a more appropriate term?

iii) Page 4, line 129. By "closed cortex", do the authors mean that the

actin is present throughout the cortex, without holes?

iv) Work on bead-grown actin networks (Ref. 18) has suggested that CP enhances Arp2/3 complex-based nucleation. This is not included in the model. The authors may or may not choose to include this effect, but a statement of why it is included or why not would be useful.

v) What does the color red denote in Figure 4a?

vi) For the actin network to exert pulling forces, it must be attached to the membrane. Do the authors have hypotheses about what causes the attachment (via the VCA for example)?

Reviewer #2

Description: This ms describes how actin assembly from a synthetic reconstituted system inside a vesicle depends on the level of capping protein (CP). The interesting result is that the membrane can be induced to protrude or to invaginate, depending on the level of CP.

General comments:

1. Overall, these observations of membrane bending, out and in, and their dependence on CP are a wonderful achievement. This group of investigators has been leading the field with their studies of actin assembly inside GUVs, and this paper represents one more example of their cutting-edge discoveries. The conclusions will be of great importance to a wide variety of researchers concerned with the cytoskeleton and membranes.

2. I am concerned about the nature of the results that underlie the conclusion that the concentration of CP determines protrusion vs invagination.

a. First, vesicles contain only one type of domain morphology, never two (top of p. 4, lines 119-125.) Why is the population of vesicles heterogeneous in this respect? What differs between two vesicles, which were created in the same tube? Is this due to fluctuations in the amounts of CP, actin or other proteins that are in each vesicle? The solution is polymerizing during the time that the vesicles are being created, so perhaps there are significant fluctuations in the results? One would like to know how much the biochemical composition of each component varies among the vesicle population, and then how this correlates with the type of domain (protrusion vs invagination).

b. Second, the following paragraph, lines 126 and onward, raises a similar concern, with the finding that protrusions "spike" were present only at 40 nM, while invaginations "bent" were present at all concentrations. Those observations do not lead me to conclude that the protrusion / invagination difference depends strongly on the CP concentration.

3. The experiments would lead to far stronger conclusions if they included a time course of observations of the vesicles, instead of one observation at the end of the process. At many points, the manuscript describes the observed results using words that give the impression of events over time. If these experiments are not possible, then these conclusions about events over time should be labeled more clearly as interpretation and speculation, extrapolating from the single-time observations.

For example, the actin accumulations at the cortex - do they form as heterogeneous, unconnected foci from the start or do they form as one homogenous layer, which then breaks into pieces? The interpretation of the occurrence of fission events would be far more convincing in a time course experiment, too. I understand that the foci in the interior contain both actin and membrane markers, and so I can imagine that fission occurred. However, visualizing it directly over time would be more convincing.

4. I recommend deleting the CP overexpression experiment section. One reason is that this section, and the paper in general, lumps different sorts of endocytosis into one. However, they differ quite a bit. Here, what is being observed is most likely the end-product of macropinocytosis, which is the engulfment of large amounts of extracellular fluid by the action of ruffles. Ruffles are very large structures, very much larger than the clathrin-mediated endocytic vesicles that form at

the plasma membranes of yeast and animal cells. Both types of structures involve and depend on actin, but in very different ways, with completely different mechanisms.

The well-known studies of how endocytosis depends on actin have been concerned with clathrin-mediated vesicle formation, in both yeast and animal cells, not macropinocytosis. See, for example, Ref 31 of the manuscript. As presented, the general reader will get the wrong impression and confuse the two processes, I am afraid.

With regard to ruffles, these particular cultured cancer cells (B16 melanoma) are misleading in that they have excessively high ruffling activity, presumably associated with their oncogenic mutations. This feature was one reason why Mejillano et al (Ref 17) used these cells in their experiments to assess the effects of knockdown of CP.

Another reason is that these experiments, in this manuscript, do not vary the CP concentration up and down - only up from endogenous. Also, the concentrations of CP that are achieved are not measured by any sort of experiment. The design is too uncontrolled to allow the interpretation offered.

Specific comments:

1. line 25: "amount OF actin"

2. line 36: do you have a citation for the Brownian ratchet model? In particular, one that describes its general acceptance? My understanding was that it was one proposed model, not the only one currently considered as valid.

3. line 50: I suggest not saying "bundles" here. Arp2/3 created branches, and the shift is to unbranched filaments. Whether unbranched filaments created bundles involves other considerations and other actin-binding proteins.

4. Some pieces of text in the Results should be in the Discussion section, in my view. These generally end paragraphs or parts of the Results section. One case is lines 136-141. These sentences are an interpretation of the results, not a description.

5. Comments above (General 2b) reveal the potential confusion of nomenclature in the manuscript. I suggest the authors choose one set of terms, define them explicitly at the outset and use them consistently throughout. In particular, I suggest "protrusion" over "spike"; the shape of the protrusions is not described by the term "spike", as I understand it. In addition, the term "bent" does not clearly indicate the direction of bending. "Bent" could refer to both protrusions and invaginations, I believe. My suggestion is "protrusion" and "invagination" but I can envision others.

6. Line 248. The types of myosins are very different, and I do not think this is merely semantics. In the yeast experiments, the myosin is class-I, not conventional class-II. Yeast have class-II conventional myosin, and it has no role in endocytosis. In these studies, the myosin is conventional, myosin-II, bipolar and contractile. I think that this type of myosin is the type that one wants for these studies, but I would simply state that it provides contractile force, pulling actin filaments together. I would not draw the parallel to myosin-based endocytosis in cells.

7. Line 374. The choice of the beta 1 isoform was unfortunate. Beta1 is found only at the Z-line of the sarcomeres of striated muscle cells. All cells, nonmuscle and muscle, contain beta2 and the beta2 isoform is the one that is found at membranes. Beta1 has some special biochemical property, which is not understood, directs it to the Z-line of sarcomeres. Admittedly, the two isoforms bind actin equally well, at least based on what has been done.

I realize that this choice was made by the authors long ago, and now many experiments have been done. I would request or suggest that one repeat any of the experiments now. However, I hope the authors will consider switching to beta2 for future work.

8. The end of the Discussion, lines 255-257, appropriately notes that regulation of CP in cells is likely to influence how the results here might apply to cells. Another consideration, in my view, is the concentration of actin, which I believe is about 100X greater in cells than in this reconstitution system. This consideration applies to essentially all biochemical experiments using actin, but it is one that I hope that physicists will keep in mind as we extrapolate from reconstitution systems to cells.

9. Line 453: "the cells and were fixed"

10. Line 454: fluorescent "phalloidin" (not unlabeled)

We thank all the referees for their insightful comments. They all encouraged us to perform a complete new series of experiments and simulations, which resulted in a major rewriting of the manuscript. Most importantly, we added now a data set on the time series of the domain formation process and an improved simulation, which resulted in a clarification of the mechanism. We are very happy that by the criticism, input and help of the referees the manuscript improved significantly.

Reviewer #1, an expert in actin assembly at interfaces, has submitted his/her report as a PDF (attached). In the paper by Henneberg et al. the authors combine *in vitro* studies with computer simulation to study shape deformations in membrane induced by arp23 branched actin networks. The authors suggest that capping proteins play a major role in this process by controlling the density of pushing filaments and their lateral distribution along the membrane, thereby causing outward or inward deformation at low and high CPs concentrations, respectively. The authors also show the impact of myosin II on these deformation processes. The addition of myosin II enhances inward deformations and promotes endocytic events. The authors also perform *in vivo* experiments showing that CPs level affects the amount of actin found in endocytic vesicles and lamellipodia in cells.

Understanding the role of arp branched network density and structure on membrane shape deformation is an emerging subject and indeed timely. Nevertheless, the paper does not go deep enough, not in terms of the experimental work nor in terms of the proposed model. The authors provide experimental and simulation results but do not provide explanations for their findings.

We thank the referee for pointing out the importance of our studies. In our results section we do provide first the-*in vitro* experiments, simulations of the kinetic model and *in-vivo* data. Later we discuss all results in our discussion section and provide explanation for our findings there. We think that the additional performed experiments and simulations point now better the underlying mechanism regarding CP controlled polymerization kinetics out.

Moreover, the authors do not provide any spatio-temporal data which is essential for understanding the impact of actin cytoskeleton structure on membrane shape deformation mechanism. Rather, the authors provide only results at the final system stage, with no information of how these systems involved into their final steady states. This is also true for the experimental part and for the simulations.

We performed a series of new experiments to be able to present now first images of the temporal structure formation. It is very important to point out, that these experiments are an extreme challenge, as the vesicle production is "offline" and the transfer from the cDice production to the microscope needs some time. In addition the diffusive lateral and rotational movement of the vesicles prevents stable long time recording. Yet, we were able to capture the major steps, and present them now in the manuscript (Fig. 3).

As a matter of fact the simulations are kinetic in nature, and thus present the temporal of network formation at the membrane. We improved the presentation to avoid such a misunderstanding.

Thus, despite the fact that the subject itself is important, this work does not provide deep insights and novel understanding on the role of actin cytoskeleton architecture on membrane deformations mechanism. Thus, in its current form, the paper does not meet the requirements for publication in Nat. Comm.

Here are my remarks:

- 1) Regarding the experimental work: The authors do not provide any movie showing the spatio-temporal information on the dynamics of VCA localization, actin nucleation, network growth, formation of actin patches, and the emergence of inward and outward deformations.

The referee comments encouraged us to perform a complete new series of experiments to give deeper insights in the temporal evolution of domain formation. It is important to emphasize that a temporal resolution in such a complex system we present is extremely challenging as domain formation at the membrane is completed within minutes. We succeeded to perform time course measurements at different

CP concentrations and were able to observe the emergence of inward and outward deformations. We now added a new section about the time laps measurements to the result section of our manuscript. These experiments allow deeper insights into the mechanisms of domain formation and allow a more detailed interpretation of our findings.

To address the dynamics of VCA localization, we labelled VCA and observed its distribution in the absence and presence of actin polymerization. In combination with the before performed temporal evolution of domain formation we provide now deeper insights in the VCA dynamics and its influence on membrane localized actin polymerization. VCA is homogenously distributed prior polymerization, and gets up-concentrated during polymerization – actually a quite important and interesting finding, which to our knowledge has not been reported yet. Further detailed study of this diffusive process is clearly beyond the scope of the current manuscript

What are the time scales for each process? Why this information is not given? It is essential to understand the dynamics of these processes.

This is a very important point raised by the referee, which we were able to address by our new set of time laps measurements. Due to its importance we also decided to include it into the manuscript. Actin domains are formed within minutes at the membrane. Here we see, that actin nucleation is immediately initiated after encapsulation and cannot be resolved by our experimental procedures. At 120 CP nucleation is slower so that first actin domains were visible after three minutes – here we do resolve the nucleation, but the signal to noise ratio and our low time resolution (to avoid phototoxic effects) only allows the unambiguous identification of the clusters after the 3 minutes.

In their in vitro system the authors stress that the density of VCA at the membrane surface is 1200 molecules per microns², which gives a distance of ~30 nm between VCA molecules. This is a very high surface density that I would expect to result in a homogenous cortical network, regardless of CP density.

The referee's comment is correct that we work at high surface concentrations of VCA at the membrane. We work at such high surface concentrations since actin based motility of μm sized beads could be only observed at very high surface densities $<20\mu\text{m}$ (Wiesner et al., JCB (2003)). We also agree with the referee that such high surface densities can result in the formation of homogeneous cortices, which is actually the case for in our experiments at 20 nM CP. But an increased CP concentration leads to lower nucleation efficiency of actin at the membrane. Thus as soon as only a few nucleation site emerge at the membrane, polymerisation is quickly initiated there and most of the actin will sort into the nucleation sites, which leads to a local monomer depletion in the direct vicinity of the polymerizing domains. Thus nucleation efficiency is even more decreased. Domain formation finally results from the fact, that our experiments are all performed with limited actin pools, thus preventing further nucleation, and actually also limiting the size of the domains). Therefore as soon as all monomers are used up, domain growth stops and the domains cannot coalesce anymore.

We improved the discussion of these important points.

Since this is not the case, and the authors do observe actin patches, it is essential to show that VCA distribution on the membrane surface is initially homogenous.

We included a new paragraph to the result section with a new set of experiments (Fig. 2). We present now the VCA distributions in the presence and absence of actin polymerization.

If VCA distribution is indeed initially homogenous, this means that VCA clusterization occurs with time, which should greatly increase VCA concentration, and thus actin nucleation and network growth at these sites. What about actin and VCA clusterization?

We conclude from our experiments, that VCA does not cluster by itself as it has been observed for Shigga Toxin (Pezeshkian et al., ACS Nano (2017)). These observations are also consistent, that in vivo VCA does not clusters itself rather its clustering is induced by the interaction with PIP₂ and proteins containing a SH3 domain (Papayannopoulos et al., Molecular Cell (2005), Rivera et al, Current Biology (2004)).

In our in-vitro approach surface densities of VCA are already high. Thus induced VCA clustering is not required to induce localized actin polymerization. From our new data, inspired by the referee's comments, we even conclude, that actin polymerization itself leads to an upconcentration of VCA within the actin domains.

What drives this process and what are the typical timescales of this process?

As discussed above VCA clustering is induced due to actin polymerization. Thus we assume that the timescales of VCA clustering are in the same range as actin domain formation. We do have neither spatial nor temporal resolution to visualize this explicitly.

How CP concentration and myosin II impact this process? Finally, do actin polymerization impacts VCA clusterization (and/or vis-versa) ?

There are no direct interaction sites known between CP and VCA. Thus VCA clustering is not expected to affect due to the presence of CP. CP can only impact VCA clustering via the interaction of the actin network and VCA. The same holds for myosin.

Note that previous experimental and theoretical works showed that there is a feedback mechanism between actin nucleators clusterization, local surface curvature, and actin polymerization (see for instance works by Basserau (expt), Gov (theory) and others). The authors should cite these works and discuss their relevance to this work.

We cited the work in the manuscript.

Similar to the impact of CPs, the effect of myosin II motors is given but not discussed. It is unclear if and how the motors affect the dynamics of network assembly and actin clusterization. Myosin motors are also suggested to be responsible for membrane invagination and fission. What are the timescale of all these processes in comparison to systems that do not include myosin motors? Labeling the motors would greatly facilitate the analysis of the data by determining their localization at the various steps of the process. Where the motors localize?

We labelled myosin and performed time course measurements. Unfortunately, myosin labelling was not successful and created too many artefacts, which is a common problem actually in the field. In this configuration we do observe colocalisation of actin-myosin clusters, yet the total actin distribution is different due to the labelling artefacts of the myosin filaments. Therefore we decided not to pursue this any further.

2) Regarding the model.

The authors use sets of equation rates to study the impact of CP on free (pushing) actin filaments barbed ends distribution at the membrane surface. In figure 4 the authors show the results of their simulations. The authors find that at low CP the distribution of filaments is higher at the center of the patch, which leads outward protrusion, while at a high CP concentration, the density of pushing filaments is higher at the edges, which lead to inward deformation. The authors do not explain the origin of this phenomenon and do not provide clear explanation to their results. In fact, it is unclear how CPs concentration controls this process and how it leads to these distinct distribution profiles.

We performed a series of additional simulations and changed the model part according to the reviewer's comments. This resulted in a major rewriting of the analysis part and we have put now more emphasis on the role of CP and on the appearance of the different density profiles.

In order to better understand the role of CP on membrane deformation, temporal evolution of the profiles of pushing barbed ends is required.

In the simulations, we report the temporal evolution of the profiles of the barbed end density. We improved the presentation to avoid any further misunderstandings.

In addition, spatio-temporal evolution of V , V^* , B , and A with time should be also given. simulation.

We now report on the spatial-temporal evolution of B and A* and the capping and branching events. By adding these additional graphs, we feel that the mechanism is now much better accessible. We decided to report these parameters, as they are best suited to get insights into the spatio-temporal evolution of the domains.

How the size r of the system chosen affect the results of the simulation.

We now report first on a system with infinite pool of molecules, there the system size does not matter for the obtained simulation results. For the finite sized systems, the parameters do indeed matter, as the size defines the number of molecules in the system. We observe the same effect as in our 1D simulation with infinite pools and without diffusion. Changing the size of the volume will effectively be equivalent to the change of concentrations, yet as the diffusion times are different the kinetic interplay will be also affected. This does not affect the major insight, that the out of equilibrium nature of the reaction is the major source for gradient formation. The full exploration of the system's dependency on all parameters is beyond the scope of this publication.

Since the authors observe actin patches formation, including VCA diffusion/transport into their equation seems to be essential.

Motivated by the referee's comment we now included the diffusive transport of VCA at membrane into the model. It did not fundamentally affect the pattern formation dependent on CP.

Finally, while experimental work on myosin II is shown it is not addressed theoretically and should be added.

The focus of this manuscript is set on the experimental work. The simulations focus on simulating the reaction kinetics and do not include an elastic model. Modeling the myosinII activity theoretically would go well beyond the scope of this paper.

Other remarks:

- There are some typos mistakes along the text which render the reading difficult.

We corrected the typo mistakes

- Citation: here I bring few examples of errors or missing citations: For instance, ref 6-8 are aimed to cite papers that discuss propulsion of spherical objects. Ref 6 – is a theoretical paper, ref 7 does not show motion at all, and ref 8 discusses reconstitution of listeria propulsion (not spherical). Thus ref 6,7 should be replaced with correct relevant papers. Ref 17-18 discuss lamellipodia to filopodia transition in vivo. The authors should also cite in vitro works. Regarding the model (in addition to ref 1,2, the authors should also cite Carlsson AE who published numerous papers on the subject).

We changed the citations according to the suggestions of the referee.

Reviewer #2, an expert in actin self-assembly (Remarks to the Author):

In this study, the authors revealed the role of CP together with the Arp2/3 complex in generating membrane deformation in a reconstituted system in vitro. Based on their observations, they challenged their model built in vitro, by manipulating the concentration of capping proteins in a cellular context.

Overall, I think that the formation of membrane protrusions at low CP concentration and negative membrane curvatures at high CP concentration is very interesting.

However, I found that the model (Figure 4) that explains why this is happening is not really clear. The authors should better explain why local actin assembly generated by high CP concentration generates invagination whereas more homogeneous actin assembly at low CP concentration generates protrusion.

In both cases actin polymerisation is inhomogeneous. The observed deformation modes result from different polymerization patterns of actin at the membrane, which are controlled by CP. We improved the

presentation of this CP dependent mechanism in our theoretical model section to avoid any misunderstanding.

I was expected from these authors a detail model based on both kinetic of actin assembly but also on the mechanic aspect of the growing network and on membrane tension to challenge their main hypothesis.

We cannot include the mechanic interplay of the membrane with the polymerizing actin domains as this would exceed the aims of this manuscript. But we now discuss the results of our kinetic model in respect to already published excellent theoretical work about exactly this question.

In their cellular overexpression experiments, the authors will not only affect assembly by the Arp2/3 complex but also inhibits assembly by formins. Maybe the authors can also inhibit locally CP activity using carmil to see if decreasing CP concentration generates protrusion in their system.

For those studies we decided to focus only on the overexpression of CP in vivo as there is already data published where CP depletion caused an explosive formation of filopodia at the cell membrane (Meillano et al.). Using Carmil to trigger CP activity in vivo is a great suggestion but would exceed the aims of this publication.

I also predict that in the model increasing Arp2/3 concentration to a very high level should have a similar effect that high CP concentration. The authors should maybe generate a phase diagram varying CP and arp2/3 complex concentration for a defined actin concentration and determine how membrane deformation is affected.

We indeed performed Arp2/3 concentration series. Here we found that Arp2/3 is important to control the membrane localized actin polymerization rather than any shape deformations of the membrane. Actin was fully localized at the membrane only at an intermediate Arp2/3 concentration (30nM). Whereas at a lower concentration of Arp2/3 (10nM) membrane localized polymerization was not very efficient, only weak membrane binding could be observed and most of the actin was still present in the vesicle's bulk. At higher Arp2/3 concentrations (450 nM) actin domain formation was still observable at the membrane but also dense actin clusters formed inside the vesicles near the membrane. Shape deformations of the membrane could be only detected at the intermediate concentration of Arp2/3 (300 nM). Here domain formation was more sensible to variation of the CP concentration. To get a full phase diagram is way beyond the scope of the manuscript- We want again emphasis that the presented experiments are extremely challenging and the ability to control structure formation within a vesicle is unique.

Reviewer #3, an expert in actin regulation including capping protein, reviewed this work together with a colleague with expertise in modelling. Rather than collate their comments into one report, he chose to append them separately:

Reviewer #1

This manuscript treats the effects of Arp2/3-induced actin polymerization inside vesicles on the shape of the membrane. It is found that increasing the concentration of capping protein shifts the membrane morphology from having regions with outward curvature to regions having inward curvature. The addition of non-muscle myosin II drives the membrane bending farther, so that vesiculation is achieved in some cases. It is proposed that this transition results from polymerization overshoots induced by increasing levels of capping, which cause a characteristic spatial profile of barbed-end density, as well as force, along the membrane. The relevance to biological cells is demonstrated via capping-protein overexpression experiments on fibroblasts, in which enhanced endocytosis is observed.

The effect of capping protein on membrane morphology demonstrated here significant and provocative, especially in view of the experiments on cells. This effect to be relevant to a broad range of actin-induced membrane-bending phenomena. However, there are problems with the mathematical modeling:

- i) The force plots in Fig 4c suggest a large net force acting on the actin network, on the order of $(200\text{pN}/\mu\text{m}^2) \times (3\ \mu\text{m}^2) = 600\ \text{pN}$. Such a large force would lead to very rapid motion of the actin network. As pointed out previously (Hassing, Julian E., et al. "Design principles for robust vesiculation in clathrin-mediated endocytosis," *Proceedings of the National Academy of Sciences* 114.7 (2017): E1118-E1127), the distribution of forces involves a balance of pulling and pushing forces so that the total force exerted by the actin network is close to zero. The force model should be revised to reflect this force balance.
- ii) It is also not clear that the increased density of free barbed ends near the edges of the actin patch, obtained at high CP concentrations, will lead to pulling forces near the center.

We follow a Brownian ratchet model. Thus, the protrusion speed of the network cannot exceed polymerization speed and equals it for high plus end density. The estimated pressure based on the plus end density is indeed up to about $500\text{pN}/\mu\text{m}^2$.

In our system membrane deformations are not created by the balance of pulling and pushing forces. The membrane deformation results from different protrusion velocities of the actin network due to an inhomogeneous barbed end distribution at the leading edge of the network at the membrane (Mogilner et al., *Biophys. J.* 83, 1237-1258 (2002)). We improved the presentation of the manuscript to avoid any misunderstanding

The proposed mechanism requires more rapid polymerization at the edges, so that the central region will be dragged back by retrograde flow. But in the absence of force, the increased density of barbed ends at the edges would cause monomer depletion (Khamviwath, Varunyu, Jifeng Hu, and Hans G. Othmer, "A continuum model of actin waves in *Dictyostelium discoideum*." *PloS one* 8.5 (2013): e64272), causing the network to grow more slowly at the edges than in the center.

It is absolutely correct that local monomer depletion in areas of high barbed end density slow down network growth and consequently the protrusion velocities. But in our system the calculated barbed end densities are lower compared to measured densities of migrating fibroblast (Vivek et al., *Biophys J.* 77, 1721-1732 (1999)). In our regime we expect that the protrusion velocities are not dominated by monomer depletion rather by membrane resistance, which damps the protrusion velocity of the actin network (Mogilner et al., *Biophys. J.* 83, 1237-1258 (2002)). Here the velocity is given by the barbed end density, and the velocity gradients are the cause for the different observed membrane deformations.

The inclusion of force would enhance this effect, since the pushing forces at the edges would slow polymerization, while the pulling forces in the center would speed polymerization. So at first glance the "overshoot" mechanism seems to give the wrong sign for the membrane bending.

It is not clear for us to what exact process the reviewer refers too. As mentioned above we don't expect that the protrusion velocity of the actin network in studied system is controlled by the force balance of pushing and pulling forces.

There are also some minor points/suggestions:

ii) The authors use the terminology "spikes" for the actin morphology at low CP concentrations, but I do not see spikes. Is there a more appropriate term?

According to the referee's comment we changed the terminology from spikes to protrusions.

iii) Page 4, line 129. By "closed cortex", do the authors mean that the actin is present throughout the cortex, without holes?

By closed cortices we refer to actin

iv) Work on bead-grown actin networks (Ref. 18) has suggested that CP enhances Arp2/3 complex-based nucleation. This is not included in the model. The authors may or may not choose to include this effect, but a statement of why it is included or why not would be useful.

Ref 18 assumes a 10x-100x higher surface density of VCA. This causes the system to work in the diffusion limit for all constituents (perfect absorber equation). We do not consider our system (or living cells) to be in this high VCA regime.

v) What does the color red denote in Figure 4a?

The color red denotes the barbed ends of the actin filaments. We commented that in the figure's caption to avoid any misunderstanding.

vi) For the actin network to exert pulling forces, it must be attached to the membrane. Do the authors have hypotheses about what causes the attachment (via the VCA for example)?

From in vitro reconstitution of *Listeria* propulsion or actin based motility at VCA coated beads we know that the network are bound to the surface (Gerbál et al., Eur Biophys J. 29, 132-140; Marcy et al., PNAS 101 5992-5997 (2004)). We expect, that the interaction of the VCA with the actin network via the Arp2/3 complex is responsible for that binding.

Reviewer #2

Description: This ms describes how actin assembly from a synthetic reconstituted system inside a vesicle depends on the level of capping protein (CP). The interesting result is that the membrane can be induced to protrude or to invaginate, depending on the level of CP.

General comments:

1. Overall, these observations of membrane bending, out and in, and their dependence on CP are a wonderful achievement. This group of investigators has been leading the field with their studies of actin assembly inside GUVs, and this paper represents one more example of their cutting-edge discoveries. The conclusions will be of great importance to a wide variety of researchers concerned with the cytoskeleton and membranes.

2. I am concerned about the nature of the results that underlie the conclusion that the concentration of CP determines protrusion vs invagination.

a. First, vesicles contain only one type of domain morphology, never two (top of p. 4, lines 119-125.) Why is the population of vesicles heterogeneous in this respect? What differs between two vesicles, which were created in the same tube? Is this due to fluctuations in the amounts of CP, actin or other proteins that are in each vesicle?

This is indeed an important finding, which also surprised us quite a bit. We attribute this to fluctuations of either concentrations of encapsulated proteins or temporal and spatial fluctuations within a vesicle resulting in the different number of nucleation sites. We find it very interesting that the number of clusters and observed morphologies correlates significantly. The probability of creating a certain number of seeds can be shifted by CP but not fully controlled, which results in the observation of the heterogeneous vesicles. In the present vesicle system, 5-6 different proteins are encapsulated, and thus even low protein fluctuations can lead to the observed fluctuations of network formation.

The solution is polymerizing during the time that the vesicles are being created, so perhaps there are significant fluctuations in the results?

To prevent spontaneous polymerization we produced the vesicles in the cold room at 4 °C, yet clearly spontaneous polymerisation cannot fully be excluded. An important experimental detail is that during the course of the experiments different actin charges have been used, which is known to introduce a variability in spontaneous nucleation, independent on the care put into the preparation.

One would like to know how much the biochemical composition of each component varies among the vesicle population, and then how this correlates with the type of domain (protrusion vs invagination).

To determine the encapsulation efficiency of the different proteins, we encapsulated Fluorescindextran (FD) with molecular sizes comparable to the encapsulated proteins. We find that the distribution of the mean intensity is rather homogeneous among the vesicles and shows low variations, which can be best seen in the attached data. As mentioned above we expect that already low fluctuations of the protein concentration are sufficient to lead to fluctuations in network formation.

b. Second, the following paragraph, lines 126 and onward, raises a similar concern, with the finding that protrusions “spike” were present only at 40 nM, while invaginations “bent” were present at all

concentrations. Those observations do not lead me to conclude that the protrusion / invagination difference depends strongly on the CP concentration.

We now clarified better why we do see the distributions of deformations in a vesicle population. As discussed before in section 2a, the distributions are due to small fluctuations of protein concentrations.

3. The experiments would lead to far stronger conclusions if they included a time course of observations of the vesicles, instead of one observation at the end of the process. At many points, the manuscript describes the observed results using words that give the impression of events over time. If these experiments are not possible, then these conclusions about events over time should be labeled more clearly as interpretation and speculation, extrapolating from the single-time observations.

For example, the actin accumulations at the cortex - do they form as heterogeneous, unconnected foci from the start or do they form as one homogenous layer, which then breaks into pieces?

The referee comments encouraged us to perform a complete new series of experiments to give deeper insights in the temporal evolution of domain formation. It is important to emphasize that a temporal resolution in such a complex system we present is extremely challenging as domain formation at the membrane is completed within minutes. We succeeded to perform time course measurements at different CP concentrations and were able to observe the emergence of inward and outward deformations. We now added a new section about the time laps measurements to the result section of our manuscript. These experiments allow deeper insights into the mechanisms of domain formation and allow a more detailed interpretation of our findings.

The interpretation of the occurrence of fission events would be far more convincing in a time course experiment, too. I understand that the foci in the interior contain both actin and membrane markers, and so I can imagine that fission occurred. However, visualizing it directly over time would be more convincing. As mentioned above: We labelled myosin and performed time course measurements. Unfortunately, myosin labelling was not successful and created too many artefacts, which is a common problem actually in the field. We could observe colocalization of actin-myosin clusters, which did not interact with the membrane at all.

4. I recommend deleting the CP overexpression experiment section. One reason is that this section, and the paper in general, lumps different sorts of endocytosis into one. However, they differ quite a bit. Here, what is being observed is most likely the end-product of macropinocytosis, which is the engulfment of large amounts of extracellular fluid by the action of ruffles. Ruffles are very large structures, very much larger than the clathrin-mediated endocytic vesicles that form at the plasma membranes of yeast and animal cells. Both types of structures involve and depend on actin, but in very different ways, with completely different mechanisms.

The well-known studies of how endocytosis depends on actin have been concerned with clathrin-mediated vesicle formation, in both yeast and animal cells, not macropinocytosis. See, for example, Ref 31 of the manuscript. As presented, the general reader will get the wrong impression and confuse the two processes, I am afraid.

With regard to ruffles, these particular cultured cancer cells (B16 melanoma) are misleading in that they have excessively high ruffling activity, presumably associated with their oncogenic mutations. This feature was one reason why Mejillano et al (Ref 17) used these cells in their experiments to assess the effects of knockdown of CP.

Another reason is that these experiments, in this manuscript, do not vary the CP concentration up and down - only up from endogenous. Also, the concentrations of CP that are achieved are not measured by any sort of experiment. The design is too uncontrolled to allow the interpretation offered.

Since referees #2 and #3.1 found the *in vivo* experiments with CP overexpression in B16-F1 cells relevant and interesting we prefer not to delete this section. This notwithstanding, to clarify the doubts that have been raised by this referee we performed additional experiments. One criticism relates to the unknown quantity of CP in the overexpressing cells. Although this is impossible to measure in a given cell, we quantified the average concentration of CP before and after induction with Doxycyclin after 24 h in the entire cell population. As now shown in Fig. S5, densitometric analyses revealed a mean 2.1-fold overexpression in the CP overexpressing cells. However, given the high variability of the expression levels in the cells as assessed by immunofluorescence intensity of the HA-labelled CP alpha 1 chain, we assume that CP overexpression in the strong overexpressors (that were used for analyses) must be considerably higher.

Moreover, we do not share the view of this referee that B16-F1 cells are not well suited to study the effects of CP overexpression. On the contrary, we purposely used this cell type, because the effects on CP depletion were previously well characterized by Mejillano and colleagues (Ref 17). As shown in this study, endogenous CP localized at extreme leading edge of lamellipodia and was excluded from filopodia. Furthermore, the depletion of CP resulted in the massive formation of filopodia at the cost of lamellipodia. Thus, these cells were ideal to study the effects of CP overexpression. HA-tagged CP not only localized to the leading edge like the wild type protein, but as one would expect based on the work of Mejillano and colleagues, microspikes/filopodia were also markedly reduced in cells overexpressing CP. Thus, we are confident to monitor authentic effects caused by overexpression of CP.

A justified criticism arises from the unknown origin of the formed vesicles. To address this issue, we used CP overexpressing cells and stained them in addition to HA-tagged CP with a battery of trafficking markers. Since neither the endosomal marker Rab7, nor the lysosomal marker LAMP1 or the fluid-phase marker TRITC-dextrane were found to colocalize with the majority of these CP-containing vesicles, we can exclude the possibility that the formed vesicles are endosomes, lysosomes or vesicular structures entrapped by micropinocytosis (as now shown in Fig. S6), but arise specifically upon overexpression of CP.

Specific comments:

1. line 25: "amount OF actin"

We thank the reviewer for reading the manuscript so carefully and corrected it in the manuscript.

2. line 36: do you have a citation for the Brownian ratchet model? In particular, one that describes its general acceptance? My understand was that it was one proposed model, not the only one currently considered as valid.

The referee's comment is correct that several models exist to describe force generation due to actin polymerization. Though the Brownian Ratchet Model is to our knowledge the most common microscopic model. Recent publications claim that actin-based propulsion is a combination of the microscopic Brownian ratchet model and macroscopic elastic gel theory (Zhu et al, PLOS (2012)). Since we do not aim to discuss the relevant theories in our manuscript we deleted the brownian ratchet model from our introduction section to avoid any confusions.

3. line 50: I suggest not saying “bundles” here. Arp2/3 created branches, and the shift is to unbranched filaments. Whether unbranched filaments created bundles involves other considerations and other actin-binding proteins.

We are not sure to understand the referee’s comment. Here we cited the work of Meijllano et al., Cell (2004), where depletion of CP results in the formation of filopodia. It is known that filopodia are created from actin bundles, which is also mentioned in the cited publication.

4. Some pieces of text in the Results should be in the Discussion section, in my view. These generally end paragraphs or parts of the Results section. One case is lines 136-141. These sentences are an interpretation of the results, not a description.

Some pieces of the results part were indeed already interpretations of the results. So we changed that and put most interpretations into the discussion part. At same points we left the interpretations, as these are important to be able to follow the argumentation and experiments.

5. Comments above (General 2b) reveal the potential confusion of nomenclature in the manuscript. I suggest the authors choose one set of terms, define them explicitly at the outset and use them consistently throughout. In particular, I suggest “protrusion” over “spike”; the shape of the protrusions is not described by the term “spike”, as I understand it. In addition, the term “bent” does not clearly indicate the direction of bending. “Bent” could refer to both protrusions and invaginations, I believe. My suggestion is “protrusion” and “invagination” but I can envision others.

We agree with the reviewer that the nomenclature was not clear enough in the manuscript. We like the suggestion to use protrusion instead of spikes. We decided to exchange bent domains by concave domains to avoid any confusion.

6. Line 248. The types of myosins are very different, and I do not think this is merely semantics. In the yeast experiments, the myosin is class-I, not conventional class-II. Yeast have class-II conventional myosin, and it has no role in endocytosis. In these studies, the myosin is conventional, myosin-II, bipolar and contractile. I think that this type of myosin is the type that one wants for these studies, but I would simply state that it provides contractile force, pulling actin filaments together. I would not draw the parallel to myosin-based endocytosis in cells.

We agree that the direct comparison between endocytic and our observed vesicles in the in-vitro system might have been a bit too provocative. So we changed it in the manuscript.

7. Line 374. The choice of the beta 1 isoform was unfortunate. Beta1 is found only at the Z-line of the sarcomeres of striated muscle cells. All cells, nonmuscle and muscle, contain beta2 and the beta2 isoform is the one that is found at membranes. Beta1 has some special biochemical property, which is not understood, directs it to the Z-line of sarcomeres. Admittedly, the two isoforms bind actin equally well, at least based on what has been done.

I realize that this choice was made by the authors long ago, and now many experiments have been done. I would request or suggest that one repeat any of the experiments now. However, I hope the authors will consider switching to beta2 for future work.

Due to the referee’s comment we verified again the origin of CP. The here used CP is mouse CP ($\alpha1/\beta2$). Accordingly we changed it in the manuscript.

8. The end of the Discussion, lines 255-257, appropriately notes that regulation of CP in cells is likely to influence how the results here might apply to cells. Another consideration, in my view, is the concentration of actin, which I believe is about 100X greater in cells than in this reconstitution system. This considerations applies to essentially all biochemical experiments using actin, but it is one that I hope that physicists will keep in mind as we extrapolate from reconstitution systems to cells.

We thank the referee for that comment and we will keep that in mind for future experiments.

9. Line 453: “the cells and were fixed”

10. Line 454: fluorescent “phalloidin” (not unlabeled)

9&10: We thank the reviewer for carefully reading the materials and methods part of the manuscript. We corrected the mistakes in the manuscript.

Reviewers' comments:

Reviewer #1 (Remarks to the Author):

Capping protein controlled actin polymerisation shapes lipid membranes by Katharina Dürre, Felix C. Keber, Philip Bleicher, Fridtjof Brauns, Christian J. Cyron, Jan Faix and Andreas R. Bausch
Based on the referees' comments the paper has improved in terms of the simulations. The simulations now take into account the fact that the system is finite and that there is depletion of material with time. Moreover, the authors take into consideration the diffusion of nucleators along the membrane, and show the effect of CP on barbed end spatiotemporal distribution. The authors performed some new experiments as requested by the referee on VCA distribution in the absence of actin assembly to show that VCA is evenly distributed prior actin polymerization. They also show the clusterization of actin in the presence of increasing concentration of CP using TIRF microscopy, which is nice and in accord with a previous work by Lee et al Science 2010 (that should be cited, see below). The authors provide in Fig. 3 and sup Fig5. data of the system at 2 additional time points, but the temporal resolution is much too low (minutes) to give any clear picture of what is actually going on. Finally, although requested in my previous report, info on the actual role of myosin motors is still missing. Although the authors did not succeed in labeling the motors, I still expect the authors to address the issues I raised, and provide movies or timelapse images on the temporal evolution of membrane invagination and fission events, that the authors suggest to be driven by the motors.

Thus, while the simulations now give a better understanding on the effect of CP on direction of the protrusions formed (outer protrusions vs. inner invaginations), I still have concerns with the experimental work, which is the main focus of this work.

Here are my remarks:

1) Fig. 3 and Supp. Fig.5 of the revised ms.

It is very difficult to extract information on the dynamics of the process from the data shown in those figures which is of very low temporal resolution.

Moreover, no movies are provided, even not for the 120 nM CP for which the system evolves much more slowly in comparison to lower CP concentrations. Why the time between frames is so large? The authors should provide data with higher temporal resolution as the added data, except for showing that the system is initially homogenous which is indeed very important, does not provide information on the temporal evolution of protrusions/invaginations.

2) In Fig. S5, the authors write "the membrane protrusion was visible one minute after production of vesicles was finished and they remained stable for at least 10 minutes". What does it mean? That membrane protrusions disappear with time? what about the actin protrusion – are they also temporally stable? The authors should clarify this point.

3) The authors write in page 9. "In our experiments the growth induced network stresses alone were not sufficient to induce fission of the invaginations from the membrane in the in vitro system. By the addition of non-muscle myosin II membrane fission events are induced and endocytic-type vesicles were formed in the in vitro system".

The authors conclude that the motors promote fission at high CP concentrations. How fission occurs? What is the mechanism?

The image quality in Fig. 4 renders the interpretation of the role of myosin II very difficult and since the process is not followed with time, it is very difficult to say what processes actually occurs.

Timelapse imaging should be provided to support this conclusion as requested already in my first report (see also Point 5 below).

4) The statistics given in Fig. S10 is done on 27 (60nM CP) and 40 vesicles (120nM CP). Why is the number of vesicles analyzed so small? I suppose that many more vesicles are present in each sample. Does it mean that only a small number of vesicles are subjected to deformation? What is the percentage of deformed vs non-deformed vesicles.

This point should be addressed in the ms.

5) In my previous report I wrote " Similar to the impact of CPs, the effect of myosin II motors is given but not discussed. It is unclear if and how the motors affect the dynamics of network assembly and actin clusterization. Myosin motors are also suggested to be responsible for membrane invagination and fission. What are the timescale of all these processes in comparison to systems that do not include myosin motors? "

Although the authors did not manage to label the motors, they were still asked to address the questions I raised. I did not see any new data on this, to my great disappointment.

The authors should address these questions by providing movies or timelapse.

6) Fig. 5.

The authors show the clusterization of VCA in the experiments due to actin polymerization. The spatio-temporal evolution of V should be added in Fig.5 or in the sup Mat. and compared to the experiments.

7) Also, in Fig. 5f it looks as if the density of barbed ends is the highest at initial times and it reduces with time until steady state is reached. How come?

8) The TIRF experiments were done with KCl and MgCl₂ concentrations different than those performed in the vesicles. Any reason for that?

9) The authors find that VCA is homogenously distributed prior polymerization, and gets upconcentrated during polymerization.

In their paper from 2010 in Science, Lee et al showed that filopodia protrusion form via clustering of WASP molecules. The authors should cite this paper as a support for their findings.

They should also refer to the theoretical work of Gov, Nir S., and Ajay Gopinathan, Dynamics of membranes driven by actin polymerization, Biophysical journal (2006). This paper discusses the coupling between actin polymerization, membrane curvature, and nucleators clustering.

10) Ref 1-8 miss previous works on the transition from lamellipodia to filopodia:

The authors should cite Lee et al Science 2010, Haviv et al PNAS (2006), and other relevant works that have been done since.

11) Line. 59 – ref. 4,23. The effect of CP on the transition from lamellipodia to filopodia was studied in vitro in Haviv et al 2006. This ref. should be mentioned here.

12) Ref. 8 - I would add/replace with:

Gov, Nir S., and Ajay Gopinathan, Dynamics of membranes driven by actin polymerization, Biophysical journal (2006). This paper discusses the coupling between actin polymerization, membrane curvature, and nucleators clustering.

In the discussion the authors should discuss the relevance of this work to their findings.

More general remarks:

Captions should include info on experimental conditions, including Supp figures.

The discussion should discuss the current findings and put them in context with previous experimental and theoretical works.

Reviewer #2 (Remarks to the Author):

The authors have addressed my concerns.

Reviewer #3 (Remarks to the Author):

These comments come from the two people who worked together as Reviewers 3.1 and 3.2

From 3.1: The authors have made appropriate changes in response to the critiques. However, with regard to the monomer depletion due to the density of barbed ends, the statement in the rebuttal that "In our regime we expect that the protrusion velocities are not dominated by monomer depletion rather by membrane resistance" is not followed up in the main text. Can the authors mention this point in the main text?

From 3.2: I continue to recommend strongly the removal of the experiments with CP overexpression in cells. The authors respond with three paragraphs, addressing three points. None of the three paragraphs successfully addresses my major concern, which is that the types of endocytosis observed in the in vitro system and observed in the cells are very different and not at all comparable. In my opinion, every cell biologist with some knowledge of endocytosis would agree with that view.

Reviewers' comments:

Reviewer #1 (Remarks to the Author):

Capping protein controlled actin polymerisation shapes lipid membranes by Katharina Dürre, Felix C. Keber, Philip Bleicher, Fridtjof Brauns, Christian J. Cyron, Jan Faix and Andreas R. Bausch

Based on the referees' comments the paper has improved in terms of the simulations. The simulations now take into account the fact that the system is finite and that there is depletion of material with time. Moreover, the authors take into consideration the diffusion of nucleators along the membrane, and show the effect of CP on barbed end spatiotemporal distribution.

The authors performed some new experiments as requested by the referee on VCA distribution in the absence of actin assembly to show that VCA is evenly distributed prior actin polymerization.

They also show the clusterization of actin in the presence of increasing concentration of CP using TIRF microscopy, which is nice and in accord with a previous work by Lee et al Science 2010 (that should be cited, see below).

The authors provide in Fig. 3 and sup Fig5. data of the system at 2 additional time points, but the temporal resolution is much too low (minutes) to give any clear picture of what is actually going on.

Finally, although requested in my previous report, info on the actual role of myosin motors is still missing. Although the authors did not succeed in labeling the motors, I still expect the authors to address the issues I raised, and provide movies or timelapse images on the temporal evolution of membrane invagination and fission events, that the authors suggest to be driven by the motors. Thus, while the simulations now give a better understanding on the effect of CP on direction of the protrusions formed (outer protrusions vs. inner invaginations), I still have concerns with the experimental work, which is the main focus of this work.

Here are my remarks:

1) Fig. 3 and Supp. Fig.5 of the revised ms.

It is very difficult to extract information on the dynamics of the process from the data shown in those figures which is of very low temporal resolution.

Moreover, no movies are provided, even not for the 120 nM CP for which the system evolves much more slowly in comparison to lower CP concentrations. Why the time between frames is so large? The authors should provide data with higher temporal resolution as the added data, except for showing that the system is initially homogenous which is indeed very important, does not provide information on the temporal evolution of protrusions/invaginations.

There seems to be a major misunderstanding und under-appreciation of the complexity of the presented experimental results. Clearly, high time and lateral resolution is always a major aim of all experimental systems, yet each experimental system has its own limitations. Working with fluorescence microscopy, the bleaching and photo-toxic effects are the major limitation to the observation time, which we now better describe in the results, figure caption and materials section. The present system shows already fluorescence artefacts after 30 sec of continuous exposure time in epifluorescence. This can be partly improved by scavenger systems, which in turn have their own limitations. In our hands, these systems strongly affected the vesicle production yield, as they affect not only the osmolarity but also the dielectric constant of the solution (the latter important for cDICE). This is the reason why we left the scavenger systems out of our vesicles. The offline production at 4°C limits the start of image acquisition. We are very proud that we were able to capture the evolution of

the process in time, by acquiring snapshots as it is presented in the manuscript (and better described). Additionally, the rotational and lateral diffusion limits the tracing of the evolving structures, especially considering that the domain formation occurs spontaneously. All of this demands a 3D imaging. The presented snapshots in time in Fig. 3 are projections of z-scans in epifluorescence (we clarified this now in the figure caption).

Even if there were sophisticated ways of circumventing some of these limitations, what kind of new insights would we be able to expect? The thickness of the domains in the final state are about 0,5-0,8 microns, the lateral dimension is about 4 microns – which is distributed on the curvature of the vesicle. Considering the optical resolution – especially in z-direction, only the lateral growth could be somewhat be resolved, given that a domain were located on the equator of the vesicle.

This is the reason, why we added in the revised manuscript a series of TIRF experiments in the presence of GOX – which overcomes some of these limitations. We are happy to provide the time-resolved movies of these experiments, but we do not consider these movies to be insightful enough to be included as supplementary material movies.

These limitations are also the reason that we turned to the simulation of the kinetic equations.

Finally, we strongly disagree with the referee: we firmly believe that the provided time lapse measurements in the vesicle system, even with the low time resolution do give important insights on the dynamics of domains formation. It can be well seen, that membrane localized polymerization clearly depends on the CP concentration, as domain formation was immediately observed at 40 nM CP and clearly delayed at 120 nM CP. Moreover the shape deformations of the membrane can be clearly attributed to the membrane localized actin polymerization process.

2) In Fig. S5, the authors write “the membrane protrusion was visible one minute after production of vesicles was finished and they remained stable for at least 10 minutes”. What does it mean? That membrane protrusions disappear with time? what about the actin protrusion – are they also temporally stable? The authors should clarify this point.

We apologize, indeed the statement in the figure caption is misleading and even wrong. We thank the referee for pointing out this error.

We wanted to point out, that the three light exposures necessary to take the images do not change the stability of the actin and membrane protrusions – they remained stable during the imaging. Clearly, taking more images would lead to phototoxic effects which would ultimately alter the structures. Thus, we did not determine what the longest exposure times were, as this is not the point of our paper. We just ensured that the captured images are without imaging artefacts.

We deleted the sentence from the figure’s caption – and added a better explanation of the limitation of fluorescence microscopy to the method section.

3) The authors write in page 9. “In our experiments the growth induced network stresses alone were not sufficient to induce fission of the invaginations from the membrane in the in vitro system. By the addition of non-muscle myosin II membrane fission events are induced and endocytic-type vesicles were formed in the in vitro system”.

The authors conclude that the motors promote fission at high CP concentrations. How fission occurs? What is the mechanism?

This is indeed an interesting question, which needs to be addressed in future studies. Certainly, the referee is appropriately pointing out, that we do not address the underlying mechanism but only report the phenomenology – which is exactly what we state in the manuscript. A study of the detailed mechanism requires in our view excessive additional experimentation and is therefore beyond the scope of the current paper.

The image quality in Fig. 4 renders the interpretation of the role of myosin II very difficult and since the process is not followed with time, it is very difficult to say what processes actually occurs. Timelapse imaging should be provided to support this conclusion as requested already in my first report (see also Point 5 below).

As detailed above, the technical limitations do not allow a time-lapse imaging of the domain formation at high resolution. Clearly, a high resolution imaging would be great, but we do not see how this would help to better identify the mechanism. We therefore decided to identify the effect of CP concentration on the resulting structures. It is not the scope of the present paper to determine the microscopic dynamics of the myosin effect.

4) The statistics given in Fig. S10 is done on 27 (60nM CP) and 40 vesicles (120nM CP). Why is the number of vesicles analyzed so small? I suppose that many more vesicles are present in each sample. Does it mean that only a small number of vesicles are subjected to deformation? What is the percentage of deformed vs non-deformed vesicles. This point should be addressed in the ms.

The yield of vesicles with myosin is rather low and the statistics plot in Fig S10 does provide the percentage of deformed vs. non-deformed vesicles, as does the Fig. S4 with many more vesicles without myosin.

5) In my previous report I wrote “ Similar to the impact of CPs, the effect of myosin II motors is given but not discussed. It is unclear if and how the motors affect the dynamics of network assembly and actin clusterization. Myosin motors are also suggested to be responsible for membrane invagination and fission. What are the timescale of all these processes in comparison to systems that do not include myosin motors? “

The timescale has to be fast, as we only were able to observe the final stable configurations. As detailed above we are unable to resolve the time course.

Although the authors did not manage to label the motors, they were still asked to address the questions I raised. I did not see any new data on this, to my great disappointment. The authors should address these questions by providing movies or timelapse.

We are very happy, that referee 2 and referee 3 are fully satisfied with our responses, and agree that further addressing the effect of the motor proteins goes beyond the present manuscript.

6) Fig. 5. The authors show the clusterization of VCA in the experiments due to actin polymerization. The spatio-temporal evolution of V should be added in Fig.5 or in the sup Mat. and compared to the experiments.

We added the requested data to the Supplementary Figures section (see Supplementary Fig. S12).

7) Also, in Fig. 5f it looks as if the density of barbed ends is the highest at initial times and it reduces with time until steady state is reached. How come?

The observation is correct that the barbed end densities decrease continuously. As described in the manuscript, this can be attributed to the depletion of the finite protein pools, which lead to continuously changing starting conditions. At the start of the simulation the number of available actin monomers is high, therefore high barbed end densities emerge at the membrane. But as the number of available actin monomers decreases due to the finite volume, the initial high barbed end densities are not reached any more at later simulations steps.

8) The TIRF experiments were done with KCl and MgCl₂ concentrations different than those performed in the vesicles. Any reason for that?

Both experiments were performed under the same salt conditions. We changed the description in the supplementary materials & methods section to clarify that point.

9) The authors find that VCA is homogenously distributed prior polymerization, and gets upconcentrated during polymerization. In their paper from 2010 in Science, Lee et al showed that filopodia protrusion form via clustering of WASP molecules. The authors should cite this paper as a support for their findings. They should also refer to the theoretical work of Gov, Nir S., and Ajay Gopinathan, Dynamics of membranes driven by actin polymerization, Biophysical journal (2006). This paper discusses the coupling between actin polymerization, membrane curvature, and nucleators clustering.

We thank the reviewer for pointing out the relevance of our findings to already published data, as they clearly support our findings. Motivated by the referee's comment we also shortly discuss the VCA clustering during polymerization in the discussions part and refer to the suggested publications.

10) Ref 1-8 miss previous works on the transition from lamellipodia to filopodia: The authors should cite Lee et al Science 2010, Haviv et al PNAS (2006), and other relevant works that have been done since.

We already cited the work of Lee et al under the first eight papers. We now cite also the Haviv + Groswasser paper.

11) Line. 59 – ref. 4,23. The effect of CP on the transition from lamellipodia to filopodia was studied in vitro in Haviv et al 2006. This ref. should be mentioned here.

Additionally, we now cite the Haviv + Groswasser paper.

12) Ref. 8 - I would add/replace with:

Gov, Nir S., and Ajay Gopinathan, Dynamics of membranes driven by actin polymerization, Biophysical journal (2006). This paper discusses the coupling between actin polymerization, membrane curvature, and nucleators clustering.

We added the reference to the first group of references as suggested.

In the discussion the authors should discuss the relevance of this work to their findings.

We now address this issue in the discussion.

More general remarks:

Captions should include info on experimental conditions, including Supp figures.

We added better descriptions to Fig. 3 and some supplemental figures. We leave it to the discretion of the editor if more experimental details are needed in the captions.

The discussion should discuss the current findings and put them in context with previous experimental and theoretical works.

We added the literature suggestions of the referee to the discussion's part and do now firmly believe that our discussion wraps up our results in the context of previously published results, but do not aim to write a review about this exciting field.

Reviewer #2 (Remarks to the Author):

The authors have addressed my concerns.

Reviewer #3 (Remarks to the Author):

These comments come from the two people who worked together as Reviewers 3.1 and 3.2

From 3.1: The authors have made appropriate changes in response to the critiques. However, with regard to the monomer depletion due to the density of barbed ends, the statement in the rebuttal that "In our regime we expect that the protrusion velocities are not dominated by monomer depletion rather by membrane resistance" is not followed up in the main text. Can the authors mention this point in the main text?

We agree with the referee's comment that the influence of monomer depletion was not well explained in the discussion's part and now revised the paragraph accordingly.

From 3.2: I continue to recommend strongly the removal of the experiments with CP overexpression in cells. The authors respond with three paragraphs, addressing three points. None of the three paragraphs successfully addresses my major concern, which is that the types of endocytosis observed in the in vitro system and observed in the cells are very different and not at all comparable. In my opinion, every cell biologist with some knowledge of endocytosis would agree with that view.

We took special care, that such misunderstandings are not any longer possible. We now state explicitly that upregulation of other components is necessary.